# QVHIGHLIGHTS: Detecting Moments and Highlights in Videos via Natural Language Queries

Jie Lei      Tamara L. Berg      Mohit Bansal
Department of Computer Science
University of North Carolina at Chapel Hill
{jielei, tlberg, mbansal}@cs.unc.edu

## Abstract

Detecting customized moments and highlights from videos given natural language (NL) user queries is an important but under-studied topic. One of the challenges in pursuing this direction is the lack of annotated data. To address this issue, we present the Query-based Video Highlights (QVHIGHLIGHTS) dataset. It consists of over 10,000 YouTube videos, covering a wide range of topics, from everyday activities and travel in lifestyle vlog videos to social and political activities in news videos. Each video in the dataset is annotated with: (1) a human-written free-form NL query, (2) relevant moments in the video w.r.t. the query, and (3) five-point scale saliency scores for all query-relevant clips. This comprehensive annotation enables us to develop and evaluate systems that detect relevant moments as well as salient highlights for diverse, flexible user queries. We also present a strong baseline for this task, Moment-DETR, a transformer encoder-decoder model that views moment retrieval as a direct set prediction problem, taking extracted video and query representations as inputs and predicting moment coordinates and saliency scores end-to-end. While our model does not utilize any human prior, we show that it performs competitively when compared to well-engineered architectures. With weakly supervised pretraining using ASR captions, Moment-DETR substantially outperforms previous methods. Lastly, we present several ablations and visualizations of Moment-DETR. Data and code is publicly available at https://github.com/jayleicn/moment_detr.

## 1   Introduction

Internet videos are growing at an unprecedented rate. Enabling users to efficiently search and browse these massive collections of videos is essential for improving user experience of online video platforms. While a good amount of work has been done in the area of natural language query based video search for complete videos (i.e., text-to-video retrieval [41, 42, 17]), returning the whole video is not always desirable, since they can be quite long (e.g., from few minutes to hours). Instead, users may want to locate precise moments within a video that are most relevant to their query or see highlights at a glance so that they can skip to relevant portions of the video easily.

Many datasets [13, 7, 18, 16, 31] have been proposed for the first task of 'moment retrieval' – localizing moments in a video given a user query. However, most of the datasets are reported [4, 18] to have a strong temporal bias, where more moments appear at the beginning of the videos than at the end. Meanwhile, for each video-query pair, all of the datasets provide annotations with only a single moment. In reality, there are often multiple moments, i.e., several disjoint moments in a video, that are related to a given query. For the second task of 'highlight detection', many datasets [38, 12, 36, 8] are query-agnostic, where the detected highlights do not change for different input user queries. [22, 43] are the two existing datasets that collect highlights based on user queries. However, only

35th Conference on Neural Information Processing Systems (NeurIPS 2021).

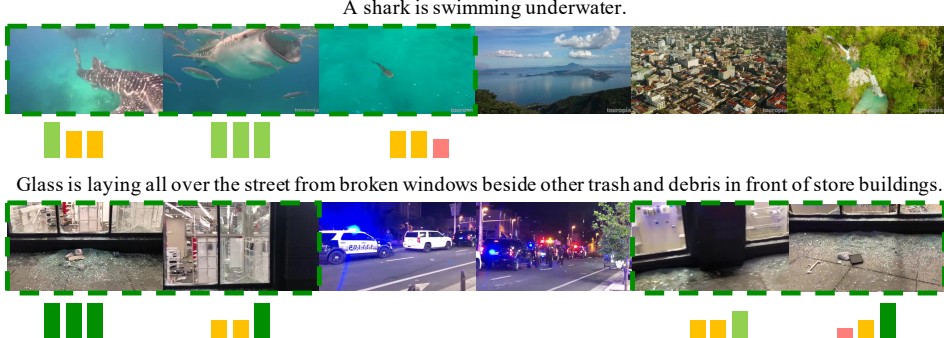

Figure 1: QVHIGHLIGHTS examples. We show localized moments in dashed green boxes. The highlightness (or saliency) scores from 3 different annotators are shown under the frames as colored bars, with height and color intensity proportional to the scores.

a small set of frames or clips are annotated ( 20 frames out of 331 seconds long videos in [22] or around 10 seconds clips out of 60 seconds video in [43]), limiting their ability to accurately learn and evaluate highlight detection methods. Lastly, although these two tasks of moment retrieval and highlight detection share many common characteristics (e.g., both require learning the similarity between user text query and video clips), they are typically studied separately, mostly due to the lack of annotations supporting both tasks in a single dataset.

To address these issues, we collect QVHIGHLIGHTS , a unified benchmark dataset that supports query-based video moment retrieval and highlight detection. Based on over 10,000 YouTube videos covering a diverse range of topics (from everyday activities and travel in lifestyle vlog videos to social and political activities in news videos), we collect high-quality annotations for both tasks. Figure 1 shows two examples from QVHIGHLIGHTS. For moment retrieval, we provide one or multiple disjoint moments for a query in a video, enabling a more realistic, accurate, and less-biased (see Section 3.2) evaluation of moment retrieval methods. Within the annotated moments, we also provide a five-point Likert-scale (from 'Very Good' to 'Very Bad') saliency/highlightness score annotation for each 2-second clip. This comprehensive saliency annotation gives more space for designing and evaluating query-based video highlight detection methods.

Next, to present strong initial models for this task, we take inspiration from recent work such as DETR [3] for object detection, and propose Moment-DETR, an end-to-end transformer encoder-decoder architecture that views moment retrieval as a direct set prediction problem. With this method, we effectively eliminate the need for any manually-designed pre-processing (e.g., proposal generation) or post-processing (e.g., non-maximum suppression) steps commonly seen in moment retrieval methods. We further add a saliency ranking objective on top of the encoder outputs for highlight detection. While Moment-DETR does not encode any human prior in its design, our experiments show that it is still competitive when compared to highly-engineered architectures. Furthermore, with additional weakly-supervised pretraining from ASR captions, Moment-DETR substantially outperforms these strong methods. Lastly, we also provide detailed ablations and visualizations to help understand the inner workings of Moment-DETR.

Overall, our contributions are 3-fold: (*i*) We collect the QVHIGHLIGHTS dataset with over 10,000 videos, annotated with human-written natural language queries, relevant moments, and saliency scores. (*ii*) We propose Moment-DETR to serve as a strong baseline for our dataset. With weakly supervised pretraining, Moment-DETR substantially outperforms several baselines, on both our proposed QVHIGHLIGHTS dataset and the moment retrieval dataset CharadesSTA [7]. (*iii*) We present detailed dataset analyses, model ablations and visualizations. For ablations, we examined various design choices of Moment-DETR as well as its pre-training strategy. We hope our work would inspire and encourage future work towards this important direction.

## 2   Related Work

**Datasets and Tasks.** Moment retrieval [13, 7, 18] requires localizing moments from a video given a natural language query. Various datasets [13, 7, 18, 16, 31] have been proposed or repurposed for

the task. However, as shown in [13, 4, 18], many of them have a strong temporal bias, where more moments are located at the beginning of the videos than the end. In Section 3.2 we show moments in QVHIGHLIGHTS distribute almost evenly over the videos. Meanwhile, while these datasets collect only a single moment for each query-video pair, we collect one or more moments. Highlight detection is another important task in our dataset. Most existing datasets [38, 12, 36, 8] are query-agnostic, which do not provide customized highlights for a specific user query. [22, 43] are the two known datasets that collect highlights based on user queries. However, they only annotate a small set of frames or clips, limiting their ability to accurately learn and evaluate highlight detection methods. In contrast, we provide a comprehensive five-point Likert-scale saliency/highlightness score annotation for all clips that are relevant to the queries. Besides, although these two tasks share some common characteristics, they are typically addressed separately using different benchmark datasets. In this work, we collect QVHIGHLIGHTS as a unified benchmark that supports both tasks. In Section 5.2 we also demonstrate that jointly detecting saliency is beneficial for retrieving moments.

**Methods.** There are a wide range of approaches developed for addressing the moment retrieval and highlight detection tasks. For highlight detection, prior methods [38, 22, 12, 21, 33] are typically ranking-based, where models are trained to give higher scores for highlight frames or clips, via a hinge loss, cross-entropy loss, or reinforcement approaches. For moment retrieval, there are work that try to score generated moment proposals [13, 35, 4, 48, 46, 40], predict moment start-end indices [9, 18, 20, 44, 47] or regress moment coordinates [7]. However, most of them require a preprocessing (e.g., proposal generation) or postprocessing step (e.g., non-maximum suppression) that are hand-crafted, and are thus not end-to-end trainable. In this work, drawing inspiration from recent work on object detection [3, 15] and video action detection [27, 45], we propose Moment-DETR that views moment retrieval as a direct set prediction problem. Moment-DETR takes video and user query representations as inputs, and directly outputs moment coordinates and saliency scores end-to-end, hence eliminating the need for any pre- or post-processing manually-designed human prior steps.

## 3 Dataset Collection and Analysis

Our QVHIGHLIGHTS dataset contains over 10,000 videos annotated with human written, free-form queries. Each query is associated with one or multiple variable-length moments in its corresponding video, and a comprehensive 5-point Likert-scale saliency annotation for each clip in the moments. In the following, we describe our data collection process and provide various data analyses.

### 3.1 Data Collection

**Collecting videos.** We would like to collect a set of videos that are less-edited and contain interesting and diverse content for user annotation. Following [6, 19], we start with user-created lifestyle vlog videos on YouTube. These are created by users from all over the world, showcasing various events and aspects of their life, from everyday activities, to travel and sightseeing, etc. These videos are captured via different devices (e.g., smartphones or GoPro) with different view angles (e.g., first-person or third-person), posing important challenges to computer vision systems. To further increase the diversity of the dataset, we also consider news videos that have large portions of 'raw footage'. These videos tend to cover more serious and world event topics such as natural disasters and protests. To harvest these videos, we use a list of queries, e.g., 'daily vlog', 'travel vlog', 'news hurricane', etc. We then download top videos that are 5-30 minutes long from YouTube's search results, keeping videos that are uploaded after 2016 for better visual quality, and filtering out videos with a view count under 100 or with a very high dislike ratio. These raw videos are then segmented into 150-second short videos for annotation.

**Collecting user queries and relevant moments.** To collect free-form natural language queries and their associated moments in the videos, we create an annotation task on Amazon Mechanical Turk. In this task, we present workers with a video and ask them to watch the video and write a query in standard English depicting interesting activities in the video. Next, we present the same worker with a grid of 2-second long clips segmented from the video, and ask them to select all clips from the grid relevant to the query. The selection can be done very efficiently via click for selecting a single clip and click-and-drag for selecting consecutive clips. This 2-second clip annotation protocol allows for more precise annotation than using 5-second clip as in [13]. Moreover, different from previous work [13, 7, 18, 16, 31] where only a single moment can be selected for a query-video pair, users can

Table 1: Top unique verbs and nouns in queries, in each video category.

| Category | #Queries | Top Unique Verbs | Top Unique Nouns |
|---|---|---|---|
| Daily Vlog | 4,473 | cook, apply, cut, clean | dog, kitchen, baby, floor |
| Travel Vlog | 4,694 | swim, visit, order, travel | beach, hotel, tour, plane |
| News | 1,143 | report, gather, protest, discuss | news, interview, weather, police |

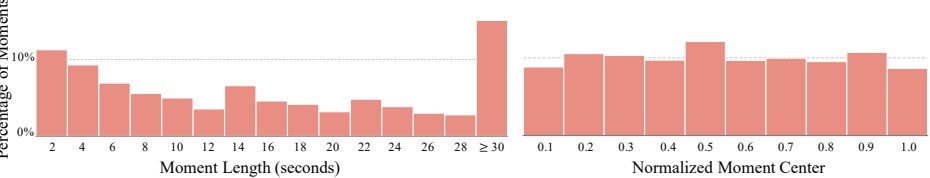

Figure 2: Distribution of moment lengths (*left*) and normalized (by video duration) center timestamps (*right*). The moments vary greatly in length, and they distribute almost evenly along the videos.

select multiple disjoint moments in our setup. To *verify* the quality of the moment annotation, we use a set of 600 query-video pairs, and collect 3 sets of moments for each query from different workers. We then calculate the Intersection-over-Union (IoU) between every pair of moments annotated for the same query, and take the average of the 3 IoU scores to check the inter-user agreement. We find that, for around 90% of queries, their moments have an average IoU score higher than 0.9, suggesting that the moments collected via our annotation process are of high inter-user agreement, thus high quality.

**Annotating saliency scores.** The relevant moment annotation in the previous step tells us which clips in the videos correspond to the user queries. Though all selected clips are relevant to the query, they may still vary greatly in their saliency, how representative they are for the query, or whether they would make a good *highlight*. For example, we would expect a clip with a proper camera angle and lighting to be better than a clip with a lot of occlusions on the activities being queried, and therefore be a better highlight for the video. Thus, we create a second annotation task targeting at collecting the saliency scores for each relevant clip. We do not ask workers to select only a small set of clips as highlights [43] because many clips may look similar and be equally salient. Hence forcing people to pick only a few clips from these similar clips can cause confusion and degrade annotation quality. Specifically, we present all the selected clips in the first task along with the queries to another set of workers. For each clip, the workers are required to rate them in a Likert-scale system[1] with five options, 'Very Good', 'Good', 'Fair', 'Bad', 'Very Bad'. As highlightness can be subjective, we collect ratings from 3 different workers and use all of them for evaluation.

**Quality control.** To ensure data quality, we only allow workers who have done more than 500 HITs and with an approval rate of 95% to participate in our annotation task. We also follow [18] to set up a qualification test. Our test contains seven multiple-choice questions (see supplementary file for an example), and workers have to correctly answer all questions in order to qualify for the task. In total, 543 workers took the test, with a pass rate of 48%. This qualification ensures high data quality – as mentioned earlier in this subsection, we observe high inter-user agreement of moment annotations. For query and saliency annotation, we pay workers $0.25 and $0.18 per query, respectively. The average hourly pay is around $11.00. The whole collection process took about 3 months and cost approximately $16,000.

### 3.2 Data Analysis

In total, we collected 10,310 queries associated with 18,367 moments in 10,148 videos. The videos are from three main categories, daily vlog, travel vlog, and news events. In Table 1, we show the number of queries in each category and the top unique verbs and nouns in the queries. These top unique words reflect the major of activities occurring in the videos. For example, in daily and travel vlog videos, the top unique verbs are mostly associated with daily activities such as 'cook' and 'clean', while in news videos, they are more about serious activities such as 'report', 'gather', 'protest'.

---

[1] https://en.wikipedia.org/wiki/Likert_scale

Table 2: Comparison with existing moment retrieval (*top*) and highlight detection (*bottom*) datasets. *Q=Query*, *MR=Moment Retrieval*, *HD=Highlight Detection*.

| Dataset | Domain | #Queries/#Videos | Avg Query Len | Avg Len (sec) Moment/Video | Avg #Moments per Query | Supported Tasks MR | HD | Has Query |
|---|---|---|---|---|---|---|---|---|
| DiDeMo [13] | Flickr | 41.2K / 10.6K | 8.0 | 6.5 / 29.3 | 1 | ✓ | - | ✓ |
| ANetCaptions [16] | Activity | 72K / 15K | 14.8 | 36.2 / 117.6 | 1 | ✓ | - | ✓ |
| CharadesSTA [7] | Activity | 16.1K / 6.7K | 7.2 | 8.1 / 30.6 | 1 | ✓ | - | ✓ |
| TVR [18] | TV show | 109K / 21.8K | 13.4 | 9.1 / 76.2 | 1 | ✓ | - | ✓ |
| YouTubeHighlights [38] | Activity | - / 0.6K | - | - / 143 | - | - | ✓ | - |
| Video2GIF [12] | Open | - / 80K | - | - / 332 | - | - | ✓ | - |
| BeautyThumb [36] | Open | - / 1.1K | - | - / 169 | - | - | ✓ | - |
| ClickThrough [22] | Open | - / 1K | - | - / 331 | - | - | ✓ | ✓ |
| ActivityThumb [43] | Activity | 10K / 4K | 14.8 | 8.7 / 60.7 | - | - | ✓ | ✓ |
| QVHIGHLIGHTS | Vlog / News | 10.3K / 10.2K | 11.3 | 24.6 / 150 | 1.8 | ✓ | ✓ | ✓ |

Table 2 shows a comparison between QVHIGHLIGHTS and existing moment retrieval and highlight detection datasets. QVHIGHLIGHTS can have multiple disjoint moments paired with a single query (on average 1.8 moments per query in a video), while all the moment retrieval datasets can only have a single moment. This is a more realistic setup as relevant content to a query in a video might be separated by irrelevant content. It also enables a more accurate evaluation since the annotations are exhaustive and clean for a single video, i.e., all the relevant moments are properly selected and no irrelevant moments are selected. In Figure 2, we show the distribution of moment lengths and normalized (by video duration) moment center timestamps. Our dataset has a rich variety of moments that vary greatly in length. Around 38% of the moments have a length of equal or less than 10 seconds, while around 23% are more than 30 seconds. The moments are almost equally distributed across the video, with a small peak in the middle (some moments span across the whole video), suggesting that our dataset suffers less from the temporal bias commonly seen in other moment retrieval datasets [13, 18] – where moments tend to occur nearer to the beginning of videos.

Most of the highlight detection datasets [38, 12, 36] in Table 2 focus on query-independent highlight detection while QVHIGHLIGHTS focuses on query-dependent highlights detection. Click-Through [22] and ActivityThumbnails [43] also have highlight annotations for queries, but their annotations are not comprehensive: for a video, ClickThrough only annotates 20 key frames and ActivityThumbnails restricts highlights to less than five clips. In contrast, we adopt a two-stage annotation process with a comprehensive 5-scale saliency score for *all* relevant clips, making it more useful for developing effective models and more accurate for evaluating model performance.

## 4 Methods: Moment-DETR

Our goal is to simultaneously localize moments and detect highlights in videos from natural language queries. Given a natural language query $q$ of $L_q$ tokens, and a video $v$ comprised of a sequence of $L_v$ clips, we aim to localize one or more moments $\{m_i\}$ (a moment is a consecutive subset of clips in $v$), as well as predicting clip-wise saliency scores $S \in \mathbb{R}^{L_v}$ (the highest scored clips are selected as highlights). Inspired by recent progress in using transformers for object detection (DETR [3]), in this work we propose a strong baseline model for our QVHIGHLIGHTS dataset, 'Moment-DETR', an end-to-end transformer encoder-decoder architecture for joint moment retrieval and highlight detection. Moment-DETR removes many hand-crafted components, e.g., proposal generation module and non-maximum suppression, commonly used in traditional methods [13, 35, 4, 48, 46, 40], and views moment localization as a direct set prediction problem. Given a set of learned moment queries, Moment-DETR models the global temporal relations of the clips in the videos and outputs moment span coordinates and saliency scores. In the following, we present Moment-DETR in detail.

### 4.1 Architecture

Figure 3 shows the overall architecture of Moment-DETR. In the following, we explain it in details.

**Input representations.** The input to the transformer encoder is the concatenation of projected video and query text features. For video, we use SlowFast [5] and the video encoder (ViT-B/32) of CLIP [30] to extract features every 2 seconds. We then normalize the two features and concatenate them at hidden dimension. The resulting video feature $v$ is denoted as $E_v \in \mathbb{R}^{L_v \times 2816}$. For query

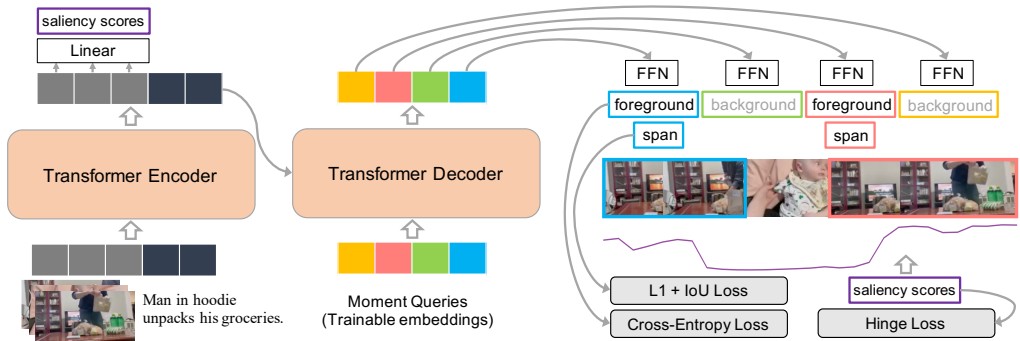

Figure 3: Moment-DETR model overview. The architecture is simple, with a transformer encoder-decoder and three prediction heads for predicting saliency scores, fore-/back-ground scores and moment coordinates. For brevity, the video and text feature extractors are not shown in this figure.

text, we use the CLIP text encoder to extract token level features, $E_q \in \mathbb{R}^{L_q \times 512}$. Next, we use separate 2-layer perceptrons with layernorm [14] and dropout [14] to project the video and query features into a shared embedding space of size $d$. The projected features are concatenated [15] at length dimension as the input to the transformer encoder, denoted as $E_{input} \in \mathbb{R}^{L \times d}$, $L = L_v + L_q$.

**Transformer encoder-decoder.** The video and query input sequence is encoded using a stack of $T$ transformer encoder layers. Each encoder layer has the same architecture as in previous work [39, 3], with a multi-head self-attention layer and a feed forward network (FFN). Since the transformer architecture is permutation-invariant, fixed positional encodings [28, 1] are added to the input of each attention layer, following [3]. The output of the encoder is $E_{enc} \in \mathbb{R}^{L \times d}$. The transformer decoder is the same as in [39, 3], with a stack of $T$ transformer decoder layers. Each decoder layer consists of a multi-head self-attention layer, a cross-attention layer (that allows interaction between the encoder outputs and the decoder inputs), and an FFN. The decoder input is a set of $N$ trainable positional embeddings of size $d$, referred to as *moment queries*.[2] These embeddings are added to the input to each attention layer as in the encoder layers. The output of the decoder is $E_{dec} \in \mathbb{R}^{N \times d}$.

**Prediction heads.** Given the encoder output $E_{enc}$, we use a linear layer to predict saliency scores $S \in \mathbb{R}^{L_v}$ for the input video. Given the decoder output $E_{dec}$, we use a 3-layer FFN with ReLU [11] to predict the normalized moment center coordinate and width w.r.t. the input video. We also follow DETR [3] to use a linear layer with softmax to predict class labels. In DETR, this layer is trained with object class labels. In our task, since class labels are not available, for a predicted moment, we assign it a *foreground* label if it matches with a ground truth, and *background* otherwise.

### 4.2 Matching and Loss Functions

**Set prediction via bipartite matching.** We denote $\hat{y} = \{\hat{y}_i\}_{i=1}^N$ as the set of $N$ predictions from the moment queries, and $y = \{y_i\}_{i=1}^N$ as the set of ground truth moments with background $\varnothing$ padding. Note that $N$ is the number of moment queries and is larger than the number of ground truth moments. Since the predictions and the ground truth do not have a one-to-one correspondence, in order to compute the loss, we need to first find an assignment between predictions and ground truth moments. We define the matching cost $\mathcal{C}_{\text{match}}$ between a prediction and a ground truth moment as:

$$\mathcal{C}_{\text{match}}(y_i, \hat{y}_{\sigma(i)}) = -\mathbb{1}_{\{c_i \neq \varnothing\}}\hat{p}_{\sigma(i)}(c_i) + \mathbb{1}_{\{c_i \neq \varnothing\}}\mathcal{L}_{\text{moment}}(m_i, \hat{m}_{\sigma(i)}), \qquad (1)$$

where each ground truth can be viewed as $y_i = (c_i, m_i)$, with $c_i$ as the class label to indicate foreground or background $\varnothing$, and $m_i \in [0, 1]^2$ a vector that defines the normalized moment center coordinate and width w.r.t. an input video; $\hat{y}_{\sigma(i)}$ is the $i$-th element of the prediction under a permutation $\sigma \in \mathfrak{S}_N$. Note that the background paddings in the ground truth are ignored in the matching cost. With this matching cost, we follow [3, 37], using the Hungarian algorithm to find the optimal bipartite matching between the ground truth and predictions: $\hat{\sigma} = \arg\min_{\sigma \in \mathfrak{S}_N} \sum_i^N \mathcal{C}_{\text{match}}(y_i, \hat{y}_{\sigma(i)})$. Based on this optimal assignment $\hat{\sigma}$, in the following, we introduce our loss formulations.

---

[2] Following [3], we use *moment queries* to refer to decoder positional embeddings, not the text queries.

**Moment localization loss.** This loss $\mathcal{L}_{\text{moment}}$ is used to measure the discrepancy between the prediction and ground truth moments. It consists of an $L1$ loss and a generalized IoU loss [32]:

$$\mathcal{L}_{\text{moment}}(m_i, \hat{m}_{\hat{\sigma}(i)}) = \lambda_{\text{L1}}||m_i - \hat{m}_{\hat{\sigma}(i)}|| + \lambda_{\text{iou}}\mathcal{L}_{\text{iou}}(m_i, \hat{m}_{\hat{\sigma}(i)}), \quad (2)$$

where $\lambda_{\text{L1}}, \lambda_{\text{iou}} \in \mathbb{R}$ are hyperparameters balancing the two terms. The IoU loss $\mathcal{L}_{\text{iou}}$ here computes 1D temporal IoU instead of 2D box IoU as in [32, 3], but they share the same formulation.

**Saliency loss.** The saliency loss is computed via a hinge loss between two pairs of positive and negative clips. The first pair is a high score clip (with index $t_{\text{high}}$) and a low score clip ($t_{\text{low}}$) within the ground-truth moments.[3] The second pair consists of one clip ($t_{\text{in}}$) within and one clip ($t_{\text{out}}$) outside the ground-truth moments. This loss is calculated as ($\Delta \in \mathbb{R}$ is the margin):

$$\mathcal{L}_{\text{saliency}}(S) = \max(0, \Delta + S(t_{\text{low}}) - S(t_{\text{high}})) + \max(0, \Delta + S(t_{\text{out}}) - S(t_{\text{in}})). \quad (3)$$

**Overall loss.** The final loss is defined as a linear combination of the losses introduced above:

$$\mathcal{L} = \lambda_{\text{saliency}}\mathcal{L}_{\text{saliency}}(S) + \sum_{i=1}^{N}[-\lambda_{\text{cls}}\log \hat{p}_{\hat{\sigma}(i)}(c_i) + \mathbb{1}_{\{c_i \neq \varnothing\}}\mathcal{L}_{\text{moment}}(m_i, \hat{m}_{\hat{\sigma}(i)})], \quad (4)$$

where $\lambda_{\text{saliency}}, \lambda_{\text{cls}} \in \mathbb{R}$ are hyperparameters for saliency and fore/background classification loss. Following [3], we down-weight the log-probability by a factor of 10 for the background class $\varnothing$ to account for class imbalance and apply classification and moment losses to every decoder layer.

### 4.3  Weakly-Supervised Pretraining via ASR

Moment-DETR is defined using an end-to-end transformer encoder-decoder architecture, eliminating the need for any human priors or hand-crafted components. Such a model typically requires a larger-scale dataset for training to unleash its true power, which would be prohibitively expensive to acquire with human labeling. Therefore, we additionally experiment with using captions from Automatic Speech Recognition (ASR) on our videos for *weakly-supervised pretraining*. Although very noisy, ASR captions have been shown to improve performance for visual recognition and text-to-video retrieval [25, 24, 20]. Specifically, we download ASR captions from YouTube, and use these caption sentences as queries, training the model to predict their corresponding timestamps. In total, we harvest 236K caption-timestamp pairs associated with 5406 *train* videos. For pretraining, the model architecture and learning objectives are the same as in standard training, except that we remove the first term in the saliency loss (Equation 3) since the saliency score annotation is not available.

## 5  Experiments and Results

### 5.1  Experimental Setup

**Data and evaluation metrics.** We split QVHIGHLIGHTS into 70% *train*, 15% *val*, and 15% *test* portions. To evaluate moment retrieval with multiple moments, we use mean average precision (mAP) with IoU thresholds 0.5 and 0.75, as well as the average mAP over multiple IoU thresholds [0.5: 0.05: 0.95], similar to action detection in [2]. We also report standard metric Recall@1 (R@1) used in single moment retrieval, where we define a prediction to be positive if it has a high IoU (>= 0.7) with one of the ground truth moments. For highlight detection, we use mAP as the main metric. We also follow [22] to use HIT@1 to compute the hit ratio for the highest scored clip. Similar to [22], we define a clip as positive if it has a score of 'Very Good'. Since we have ground truth saliency scores from 3 users, we evaluate performance against each then take the average.

**Implementation details.** Our model is implemented in PyTorch [29]. We set the hidden size $d$=256, #layers in encoder/decoder $T$=2, #moment queries $N$=10. We use dropout of 0.1 for transformer layers and 0.5 for input projection layers. We set the loss hyperparameters as $\lambda_{\text{L1}}$=10, $\lambda_{\text{iou}}$=1, $\lambda_{\text{cls}}$=4, $\lambda_{\text{s}}$=1, $\Delta$=0.2. The model weights are initialized with Xavier init [10]. We use AdamW [23] with an initial learning rate of 1e-4, weight decay 1e-4 to optimize the model parameters. The model is trained for 200 epochs with batch size 32. For pretraining, we use the same setup except that we train the model for 100 epochs with batch size 256. Both training/finetuning and pretraining are conducted on an RTX 2080Ti GPU, with training/finetuning taking  12 hours and pretraining  2 days.

---

[3]We average the saliency scores from 3 annotators and then choose a pair of high and low score clips.

Table 3: Baseline Comparison on QVHIGHLIGHTS *test* split. We highlight the best score in each column in **bold**, and the second best score with underline. XML+ denotes our improved XML [18] model. *PT* denotes weakly supervised pretraining with ASR captions. For Moment-DETR variants, we also report standard deviation of 5 runs with different random seeds.

| Method | Moment Retrieval | | | | | Highlight Detection >= Very Good | |
| --- | --- | --- | --- | --- | --- | --- | --- |
| | R1 | | mAP | | | | |
| | @0.5 | @0.7 | @0.5 | @0.75 | avg | mAP | HIT@1 |
| BeautyThumb [36] | - | - | - | - | - | 14.36 | 20.88 |
| DVSE [22] | - | - | - | - | - | 18.75 | 21.79 |
| MCN [13] | 11.41 | 2.72 | 24.94 | 8.22 | 10.67 | - | - |
| CAL [4] | 25.49 | 11.54 | 23.40 | 7.65 | 9.89 | - | - |
| CLIP [30] | 16.88 | 5.19 | 18.11 | 7.00 | 7.67 | 31.30 | **61.04** |
| XML [18] | 41.83 | 30.35 | 44.63 | 31.73 | 32.14 | 34.49 | 55.25 |
| XML+ | 46.69 | 33.46 | 47.89 | 34.67 | 34.90 | 35.38 | 55.06 |
| Moment-DETR | 52.89 $_{\pm 2.3}$ | 33.02 $_{\pm 1.7}$ | 54.82 $_{\pm 1.7}$ | 29.40 $_{\pm 1.7}$ | 30.73 $_{\pm 1.4}$ | 35.69 $_{\pm 0.5}$ | 55.60 $_{\pm 1.6}$ |
| Moment-DETR w/ PT | **59.78** $_{\pm 0.3}$ | **40.33** $_{\pm 0.5}$ | **60.51** $_{\pm 0.2}$ | **35.36** $_{\pm 0.4}$ | **36.14** $_{\pm 0.25}$ | **37.43** $_{\pm 0.2}$ | 60.17 $_{\pm 0.7}$ |

## 5.2 Results and Analysis

**Comparison with baselines.** We compare Moment-DETR with various moment retrieval and highlight detection methods on the QVHIGHLIGHTS test split; results are shown in Table 3. For moment retrieval, we provide three baselines, two proposal-based methods MCN [13] and CAL [4], and a span prediction method XML [18]. For highlight detection, we provide two baselines, BeautyThumb [36] based solely on frame quality, and DVSE [22] based on clip-query similarity. Since XML also outputs clip-wise similarity scores to the user query, we provide highlight detection results for this model as well. The original XML model has a smaller capacity than Moment-DETR, hence for a fair comparison, we increased its capacity by adding more layers and train it for the same number of epochs as Moment-DETR. Moreover, to leverage the saliency annotations in QVHIGHLIGHTS, we further added an auxiliary saliency loss to it (referred to as 'XML+'). These enhancements improve the original XML model by 2.76 average mAP. In addition, for both tasks, we also add CLIP [30] as a baseline. Specifically, we compute clip-wise similarity scores by computing image-query scores where the image is the center frame of the clip. For highlight detection, we directly use these scores as prediction; for moment retrieval, we use TAG [49] to progressively groups top-scored clips with the classical watershed algorithm [34].

Compared to the best baseline XML+, Moment-DETR performs competitively on moment retrieval, where it achieves significantly higher scores on a lower IoU threshold, i.e., R1@0.5 and mAP@0.5 (>7% absolute improvement), but obtains lower scores on higher IoU threshold, i.e., R1@0.7 and mAP@0.75. We hypothesize that this is because the L1 and generalized IoU losses give large penalties only to large mismatches (i.e., small IoU) between the predicted and ground truth moments. This property encourages Moment-DETR to focus more on predictions with small IoUs with the ground truth, while less on those with a reasonably large IoU (e.g., 0.5). This observation is the same as DETR [3] for object detection, where it shows a notable improvement over the baseline in $AP_{50}$, but lags behind in $AP_{75}$. For highlight detection, Moment-DETR performs similarily to XML+. As discussed in Section 4.3, Moment-DETR is designed without human priors or hand-crafted components, thus may require more training data to learn these priors from data. Therefore, we also use ASR captions for weakly supervised pretraining. With pretraining, Moment-DETR greatly outperforms the baselines on both tasks, showing the effectiveness of our approach.[4] One surprising finding is that CLIP [30] gives the best highlight detection performance in terms of HIT@1, though its overall performance is much lower than Moment-DETR.

**Loss ablations.** In Table 4, we show the impact of the losses by turning off one loss at a time. When turning off the saliency loss, we observe a significant performance drop for highlight detection, and surprisingly moment retrieval as well. We hypothesize that the moment span prediction losses (L1 and IoU) and classification (CLS) loss do not provide strong supervision for learning the similarity

---

[4]We also tried pretraining with XML+, and under careful tuning, the results are still worse than without pretraining, which might because XML+'s cross-entropy loss gives strong penalties to small mismatches, preventing it from learning effectively from noisy (thus many small mismatches) data.

Table 4: Loss ablations on QVHIGHLIGHTS *val* split. All models are trained from scratch.

| L1 | gIoU | Saliency | CLS | Moment Retrieval | | | Highlight Detection (>=Very Good) | |
|---|---|---|---|---|---|---|---|---|
| | | | | R1@0.5 | R1@0.7 | mAP avg | mAP | Hit@1 |
| ✓ | ✓ | | ✓ | 44.84 | 25.87 | 25.05 | 17.84 | 20.19 |
| ✓ | | ✓ | ✓ | 51.10 | 31.16 | 27.61 | 35.28 | 54.32 |
| | ✓ | ✓ | ✓ | 50.90 | 30.97 | 28.84 | **36.61** | **56.71** |
| ✓ | ✓ | ✓ | ✓ | **53.94** | **34.84** | **32.20** | 35.65 | 55.55 |

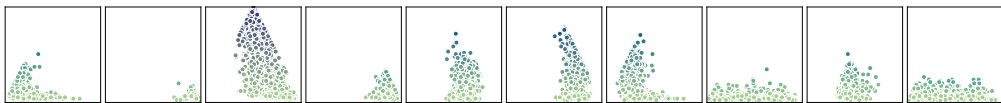

Figure 4: Visualization of all moment span predictions for all the 1550 videos on QVHIGHLIGHTS val split, for all the 10 moment query slots in Moment-DETR decoder. x-axis denotes the normalized moment span center coordinates w.r.t. the videos, y-axis denotes the normalized moment span width (also indicated by color). We observe that each slot learns to predict moments in different temporal locations and different widths. For example, the first slot mostly predicts short moments near the beginning of the videos, while the second slot mostly predicts short moments near the end.

Table 5: Effect of pretraining data domain and size, results are on QVHIGHLIGHTS *val* split.

| Pretraining Videos | Moment Retrieval | | | Highlight Detection | |
|---|---|---|---|---|---|
| | R1@0.5 | R1@0.7 | mAP avg | mAP | HIT@1 |
| None | 53.94 | 34.84 | 32.20 | 35.65 | 55.55 |
| 2.5K HowTo100M | 54.58 | 36.45 | 32.82 | 36.27 | 56.45 |
| 5K HowTo100M | 55.23 | 35.74 | 32.84 | 36.82 | 58.39 |
| 10k HowTo100M | 56.06 | 38.39 | 34.16 | 35.97 | 56.84 |
| 5.4K QVHIGHLIGHTS | **59.68** | **40.84** | **36.30** | **37.70** | **60.32** |

between the input text queries and their relevant clips, while the saliency loss gives a direct signal to learn such similarity. Under our framework, this also suggests that *jointly detecting saliency is beneficial to retrieve moments*. When turning off one of the span predictions losses (L1 or IoU), we see a notable drop in moment retrieval performance while the highlight detection performance stays similar, showing that both losses are important for moment retrieval.

**Moment query analysis.** In Figure 4, we visualize moment span predictions for all the 1550 QVHIGHLIGHTS *val* videos, for the 10 moment query slots in Moment-DETR decoder. As shown in the figure, each slot learns to predict moments of different patterns, i.e., different temporal locations and different widths. For example, some slots learn to predict short moments near the beginning or end of the videos (e.g., the first two slots), while some slots learn to predict both short and long moments near the center (e.g., the third slot). Overall, most of the slots learn to predict short moments while only a handful of them learn to predict long moments, possibly because there are more short moments in QVHIGHLIGHTS than long moments (see our data analysis in Section 3.2).

**Pretraining data domain and size.** To better understand the role of pretraining, we examine the effect of using data of different domain and sizes for pretraining. Specifically, we use HowTo100M [25] instructional videos of different sizes for pretraining Moment-DETR and then evaluate its finetuning performance on QVHIGHLIGHTS dataset. The results are shown in Table 5. We notice that out-of-domain pretraining on HowTo100M videos also improves the model performance, even when only trained on 2.5K videos. When increasing the number videos, we see there is a steady performance gain for the moment retrieval task. While for highlight detection, the performance fluctuates, probably because of the pretraining task is not well aligned with the goal of detecting highlights. Compared to in-domain and out-of-domain videos, we notice that 5.4K in-domain QVHIGHLIGHTS videos greatly outperforms 10K HowTo100M videos, demonstrating the importance of aligning the pretraining and the downstream task domain.

Table 6: Results on CharadesSTA [7] *test* split.

| Method | R1@0.5 | R1@0.7 |
|---|---|---|
| CAL [4] | 44.90 | 24.37 |
| 2D TAN [48] | 39.70 | 23.31 |
| VSLNet [47] | 47.31 | 30.19 |
| IVG-DCL [26] | 50.24 | 32.88 |
| Moment-DETR | 53.63 | 31.37 |
| Moment-DETR w/ PT (on 10K HowTo100M videos) | **55.65** | **34.17** |

An Asian woman wearing a Boston t-shirt is in her home talking.

Saliency scores

A family is playing basketball together on a green court outside.

Saliency scores

Figure 5: Prediction visualization. Predictions are shown in solid red boxes or lines, ground-truth are indicated by dashed green lines. *Top* row shows a correct prediction, *bottom* row shows a failure.

**Generalization to other datasets.** In Table 6, we test Moment-DETR's performance on moment retrieval dataset CharadesSTA [7]. Similar to our observations in Table 3, when comparing to SOTA methods, we notice that Moment-DETR shows a significant performance gain on R1@0.5, while performs slightly worse on a tighter metric, i.e., R1@0.7. Meanwhile, after pretrained on 10K HowTo100M videos, Moment-DETR's performance is greatly improved, setting new state-of-the-art for the dataset. This shows Moment-DETR's potential to adapt to different data and tasks.

**Prediction visualization.** Figure 5 (*top*) shows a correct prediction from Moment-DETR. We can see that the model is able to correctly localize two disjoint moments relevant to the user query. Meanwhile, the saliency scores also align very well with the ground truth score curve (obtained by averaging the scores from 3 annotators). And not surprisingly, this saliency score curve also matches the moment predictions – where we see higher scores for localized regions than those outside. Figure 5 (*bottom*) shows a failure case, where it incorrectly localized a partially true moment (2nd frame) where the family is playing on a court but not playing basketball. More examples in Appendix.

# 6   Conclusion

We collect QVHIGHLIGHTS dataset for moment retrieval and highlight detection from natural language queries. This new dataset consists of over 10,000 diverse YouTube videos, each annotated with a free-form query, relevant moment timestamps and clip-wise saliency scores. Detailed data analyses are provided comparing the collected data to previous works. We further propose Moment-DETR, an encoder-decoder transformer that jointly perform moment retrieval and highlight detection. We show that this new model performs competitively with baseline methods. Additionally, it also learns effectively from noisy data. With weakly supervised pretraining using ASR captions, Moment-DETR substantially outperforms previous methods, setting a strong precedence for future work. Lastly, we provide ablations and prediction visualizations of Moment-DETR.

**Social Impact** The predictions from our system reflect the distribution of the collected dataset. These predictions can be inaccurate, and hence users should not completely rely on our predictions for making real-world decisions (similar to previous work on modeling video-based predictions). In addition, please see license and usage of the data and code in the supplementary file.

**Acknowledgements:** We thank Linjie Li for the helpful discussion. This work is supported by NSF Award #1562098, DARPA KAIROS Grant #FA8750-19-2-1004, ARO-YIP Award #W911NF-18-1-0336, DARPA MCS Grant N66001-19-2-4031, and a Google Focused Research Award. The views contained in this article are those of the authors and not of the funding agency.

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
