# Appendix for
# QVHIGHLIGHTS: Detecting Moments and Highlights in Videos via Natural Language Queries

**Jie Lei**     **Tamara L. Berg**     **Mohit Bansal**
Department of Computer Science
University of North Carolina at Chapel Hill
{jielei, tlberg, mbansal}@cs.unc.edu

## A   Additional Results

**Performance breakdown by video category.** In Table 1, we show model performance breakdown on the 3 major video categories: daily vlog, travel vlog and news.

Table 1: Performance breakdown by video category, on QVHIGHLIGHTS *test* split. We highlight the best score in each column in **bold**, and the second best score with underline. All models are trained from scratch.

| Method | Moment Retrieval \| R1 IoU=0.5 | | | Highlight Detection \| HIT@1 | | |
|---|---|---|---|---|---|---|
| | daily (46.5%) | travel (43.1%) | news (10.4%) | daily (46.5%) | travel (43.1%) | news (10.4%) |
| BeautyThumb [5] | - | - | - | 24.13 | 17.44 | 20.62 |
| DVSE [4] | - | - | - | 21.90 | 21.50 | 22.50 |
| MCN [2] | 8.23 | 14.44 | 13.12 | - | - | - |
| CAL [1] | 24.83 | 26.92 | 22.50 | - | - | - |
| XML [3] | 45.05 | 40.45 | 33.12 | 58.58 | 53.08 | 49.38 |
| XML+ | 49.37 | 46.62 | 35.00 | 57.18 | 54.44 | 48.12 |
| Moment-DETR | 51.80 | 56.57 | 42.50 | 56.15 | 56.93 | 47.62 |
| Moment-DETR w/ PT | **63.22** | **59.08** | **48.63** | **60.27** | **61.95** | **51.75** |

**Ablations on #moment queries.** In Table 2, we show the effect of using different #moment queries. As can be seen from the table, this hyper-parameter has a large impact on moment retrieval task where a reasonably smaller value (e.g., 10) gives better performance. For highlight detection, the performance does not change much in terms of mAP, but HIT@1 favors smaller number of moment queries as well. Considering performance of both tasks, our best model use 10 moment queries.

Table 2: Ablations on #moment queries on QVHIGHLIGHTS *val* split.

| #Moment Queries | Moment Retrieval | | | Highlight Detection (>=Very Good) | |
|---|---|---|---|---|---|
| | R1@0.5 | R1@0.7 | mAP avg | mAP | Hit@1 |
| 5 | **54.90** | 34.06 | 31.08 | 36.04 | **57.03** |
| 10 | 53.94 | **34.84** | **32.20** | 35.65 | 55.55 |
| 20 | 47.94 | 29.10 | 24.81 | **36.34** | 55.94 |
| 50 | 39.81 | 21.16 | 18.47 | 34.96 | 53.48 |
| 100 | 41.16 | 21.68 | 19.51 | 34.52 | 51.87 |

**Saliency loss ablations.** As described in main text Equation 3, Moment-DETR's saliency loss consists of two terms, one term that distinguishes between high and low score clips (i.e., $t_{\text{high}}$, $t_{\text{low}}$),

another term distinguishes between clips in and outside the ground-truth moments (i.e., $t_{in}$, $t_{out}$). In Table 3, we study the effect of using the two terms. We notice that adding one of them improves the model performance across all metrics, while the term ($t_{in}$, $t_{out}$) typically works better. Overall, the best performance is achieved by using both terms.

Table 3: Ablations on saliency loss on QVHIGHLIGHTS *val* split.

| Saliency Loss Type | Moment Retrieval | | | Highlight Detection (>=Very Good) | |
|---|---|---|---|---|---|
| | R1@0.5 | R1@0.7 | mAP avg | mAP | HIT@1 |
| None | 44.84 | 25.87 | 25.05 | 17.84 | 20.19 |
| ($t_{in}$, $t_{out}$) | 52.90 | **36.32** | 31.46 | 35.62 | 52.58 |
| ($t_{high}$, $t_{low}$) | 52.52 | 33.16 | 30.35 | 29.32 | 40.77 |
| ($t_{in}$, $t_{out}$) + ($t_{high}$, $t_{low}$) | **53.94** | 34.84 | **32.20** | **35.65** | **55.55** |

**More prediction examples.** We show more correct predictions and failure cases from our Moment-DETR model in Figure 1 and Figure 2.

## B  Additional Data Analysis and Collection Details

**Distribution of saliency scores.** In Table 4, we show the distribution of annotated saliency scores. We noticed 94.41% of the annotated clips are rated by two or more users as 'Fair' or better (i.e., >=3, meaning they may be less saliency, but still relevant, see supplementary file Figure 6). Only 0.96% of the clips have two or more users rated as 'Very Bad'. This result is consistent with our earlier moment verification experiments.

Table 4: Distribution of annotated saliency scores on QVHIGHLIGHTS *train* split. Since we have scores from 3 users, we show the percentage as two or more users agree on a certain setting, e.g., at least two users agree that 5.59% of the clips should be rated with a score lower than or equal to 'Bad'.

| Score | =1 (Very Bad) | <=2 (Bad) | <=3 (Fair) | <=4 (Good) | <=5 (Very Good) |
|---|---|---|---|---|---|
| Percentage of Clips | 0.96 | 5.59 | 23.44 | 62.10 | 100.00 |

**Annotation Instructions and Interfaces.** To ensure data quality, we require workers to pass our qualification test before participating in our annotation task. We show an example question from our qualification test in Figure 3. Our data collection process consists of two stages: (1) query and moment annotation, we show its instructions and annotation interface in Figure 4 and Figure 5, respectively; (2) saliency score annotation, we show its instructions and interface in Figure 6.

## C  Content, License and Usage.

Our data[1] and code[2] are publicly available at `https://github.com/jayleicn/moment_detr`. Additionally, this dataset should be used for research purposes only and not be used for any purpose (e.g., surveillance) that may violate human rights. The videos in the dataset are collected from a curated list of non-offensive topics such as vlogs and news. We use these YouTube videos under the Fair Use.[3] Our study was conducted on Amazon Mechanical Turk (AMT), based on an IRB application approved by our university IRB officials. The collected data via AMT does not contain any personally identifiable information.

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

Some friends have a birthday meal together around a large table at a restaurant.

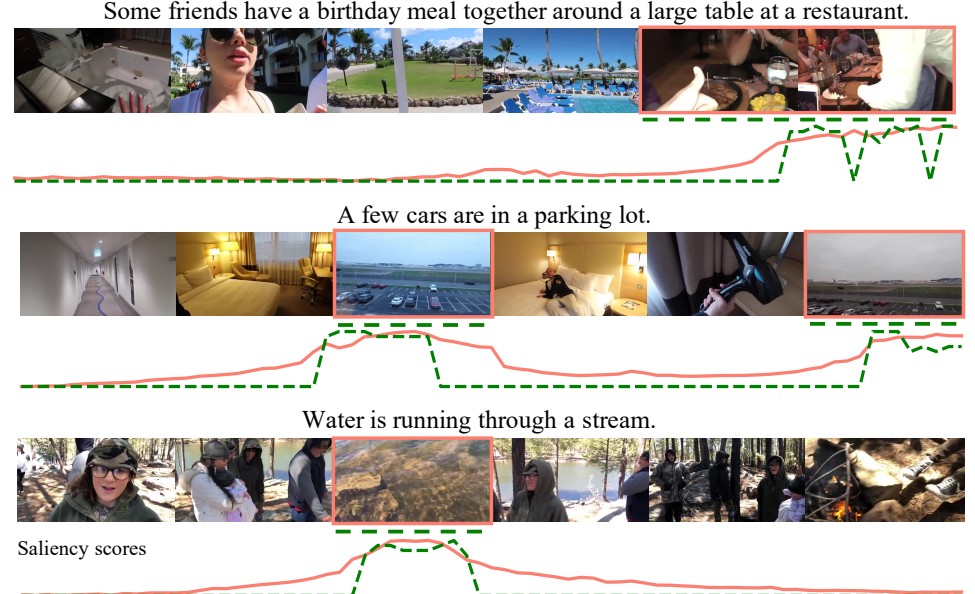

A few cars are in a parking lot.

Water is running through a stream.

Saliency scores

Figure 1: Correct predictions from Moment-DETR. Predictions are shown in solid red boxes or lines, ground-truth are indicated by dashed green lines.

A man cuts watermeoln into small peices on a glass tray.

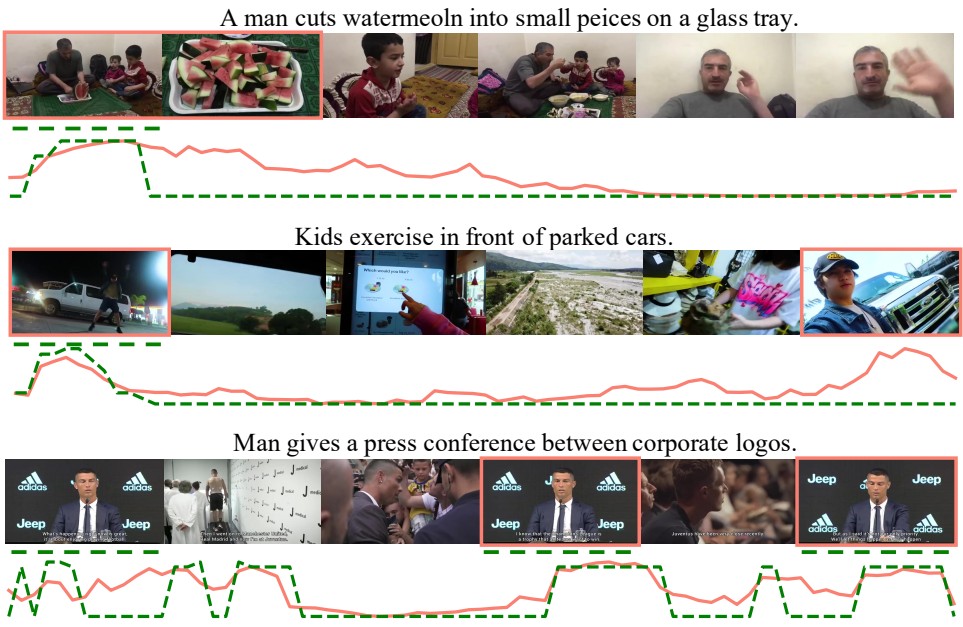

Kids exercise in front of parked cars.

Man gives a press conference between corporate logos.

Figure 2: Wrong predictions from Moment-DETR. Predictions are shown in solid red boxes or lines, ground-truth are indicated by dashed green lines.

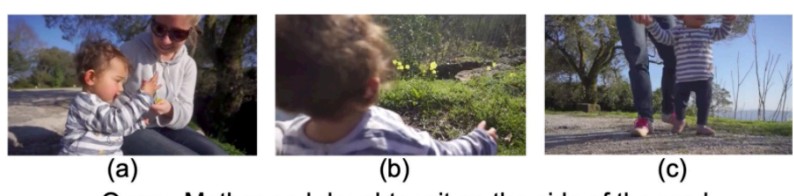

Query: Mother and daughter sit on the side of the road.

| Question: | Based on the query, which one is the best as a highlight for the video? |
|---|---|

○ A  (a)

○ B  (b)

○ C  (c)

Figure 3: Example question from our qualification test.

# Write a query and locate it in the video.

Detailed Instructions & Examples (click to show/hide)

You will be showing a 2-3 minutes long video. Your task is to find and write a sentence that describes a salient visual event in the video, and select all video segments/clips related to this event.

**Steps:**

1. ↻ Click to watch the video, then write a sentence (aka. **event query**) to describe one of the main events in the video.
   - An event can be:
     - activities of people or animal, for example,  A man in blue top is surfing.
     - or anything else that are **visually salient and important** in the video. While describing what you hear from the video is also acceptable, we encourage you to always **describe events that can be seen**.
   - It should be relevant to >=10% of the video segments. After hitting the submit button, our system will automatically notify you if it is too short. Note that the whole video can be relevant to your query, in which you need to select all the segments.
   - The description should be **a single sentence** written in standard English, and contains at least 5 words.
   - Be specific, **avoid general and boring ones** like "Two people are talking" (✗), "people vlog their day" (✗), "adventure to a hotel" (✗).
   - **Write different events for different videos**, repetitions should be avoid.
   - Please describe events located at different parts of the videos, that is, **do not always describe events at the beginning**.
   - Some videos are in **foreign languages**, you can describe what is happening from what you see.

2. Select **all** the video segments below that are **relevant to this event**.
   - The long video is split into 2-seconds long segments and are shown in the selection area below.
   - **Select all query-relevant video segments**. They can be consecutive or non-consecutive.

Examples with explanations (**Read/Understand all of them helps you to get a higher approval rate!**):

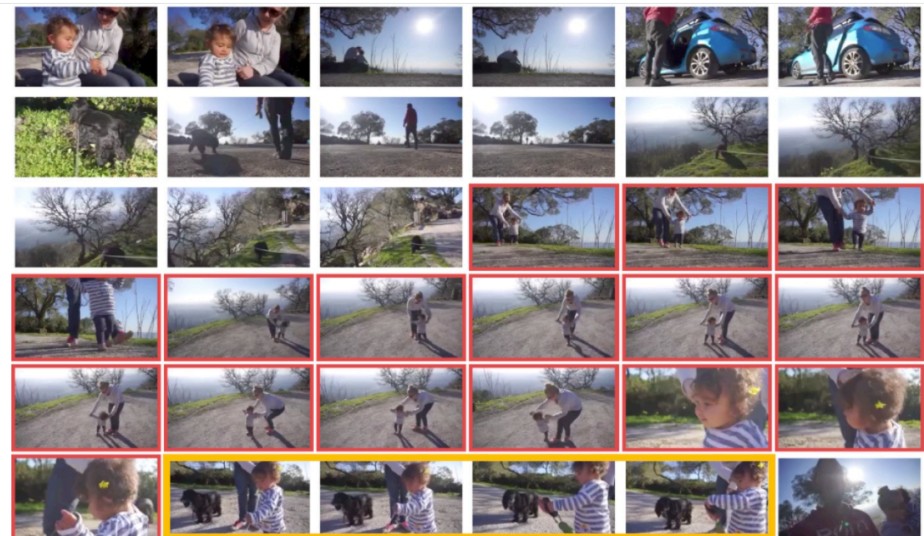

Query-relevant clips/segments are circled by red-boxes. Other symbols are used in the explanations.

Event Query: Mother tries to help daughter walk down the sidewalk.

| Event Explanation: | correct | Good! It describes a visually salient event with some details. |
|---|---|---|
| Relevant Explanation: | correct | Good! The selected clips (by red boxes) all match the event query, and no irrelevant clips are selected. Note that the few clips circled by the thick yellow boxes should not be selected, as they have stopped walking and started playing with dogs. |

You may need to scroll inside the text boxes above to see all the text.

Last example  **1 / 4**  Next example

Figure 4: Annotation instructions (with some examples) for collecting queries and moments.

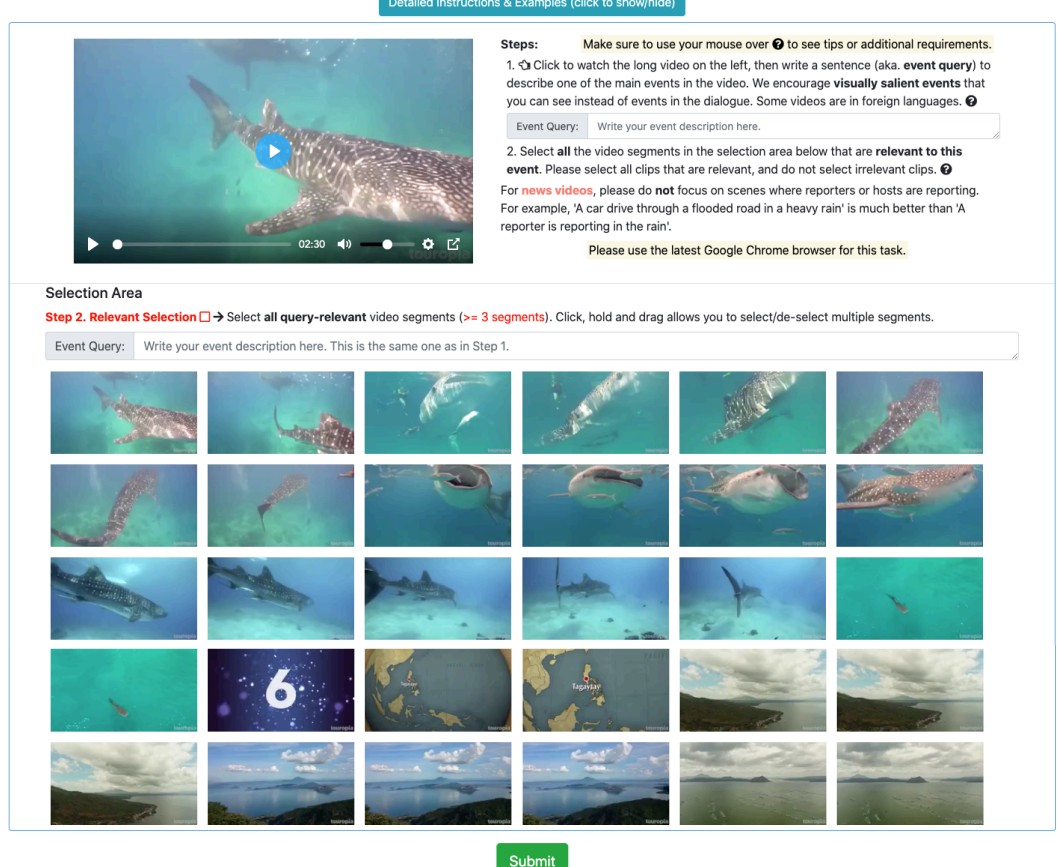

Figure 5: Annotation interface for collecting queries and moments. A short version of the instruction is also shown, the complete instructions (shown in Figure 4) can be viewed by clicking the top green button *Detailed Instructions & Examples (click to show/hide)*.

Figure 6: Annotation instructions and interface for saliency score annotation.