# OpenReview forum: "Detecting Moments and Highlights in Videos via Natural Language Queries"
_NeurIPS.cc/2021/Conference — NeurIPS 2021 Poster_

### Official Review · Reviewer_zVQi · 2021-07-10

**Rating:** 5
**Confidence:** 5

**Summary:**

This paper introduces a new dataset using language queries to localize moments in videos. The paper also introduces a DETR-like baseline for this task. Several existing models and the baseline are run on the proposed dataset. I'm concerned the contributions will not be significant, the dataset is small (10k videos) and the baseline is an adaptation of existing methods to videos.

**Ethical Concerns:**

The paper does raise a few potential ethical concerns, regarding the dataset. The dataset consists of "over 10,000 YouTube videos, covering a wide range of topics, from everyday activities and travel in lifestyle vlog videos to social and political activities in news videos"

This is concerning because it can provide annotations that could be used to violate privacy (e.g., using personal vlog videos) or enabling other agencies to use this data to build models to understand vlog-style videos for undesirable purposes. Also having societal and political activities (e.g., protests as mentioned in the paper) in the data could be used for undesirable surveillance purposes.  Given that there are protests, there could also be violent imagery in this dataset (not mentioned if this has been removed or not).


**Ethics Review Area:**

["Inadequate Data and Algorithm Evaluation", "Inappropriate Potential Applications & Impact  (e.g., human rights concerns)"]

**Limitations And Societal Impact:**

There is some discussion on the limitations of the proposed pretraining, but otherwise there is no discussion on the limitations of the approach or dataset and no meaningful discussions on the societal impacts of the dataset.

**Main Review:**

Overall the paper is fairly well written, and easy to understand. The task of localizing moments in videos from language queries is interesting, important and would be quite valuable to solve. The code and dataset annotations are provided, which should help in reproducing and extending the work.

However, the paper has a few weakness currently:

(1) Throughout the paper, it is mentioned that the dataset is diverse, but I do not see any metrics of diversity of the dataset. It seems misleading to claim diversity without providing any support for it. Also, the paper does not have any mention of ethical concerns or societal impact. Given that such a dataset was created, especially using personal VLOG style videos, this seems like it should have some discussion.

(2) The dataset is also fairly small by current standards, 10,000 videos is not many.

(3) The proposed baseline approach is fairly straightforward adaptation of existing approaches, like DETR to videos. There is some value in seeing it work and providing a good baseline for the new dataset. It would also be good to compare the proposed baseline to previous works on other, existing datasets to help make comparisons.

In the checklist:
"Did you include the estimated hourly wage paid to participants and the total amount spent on participant compensation? [Yes] See Section 3.1."

I do not see any estimated payments, wage, or time spent on the task in Section 3.1 or the paper or supplementary materials.

**Needs Ethics Review:**

Yes

**Time Spent Reviewing:**

2

---

> ### Author Response · Authors · 2021-08-10
> **Author Response (zVQi)**
>
> Thanks for reviewing our work and giving constructive feedback. We answer your questions below:
>
> ---------------
>
> > (1) Throughout the paper, it is mentioned that the dataset is diverse, but I do not see any metrics of diversity of the dataset. It seems misleading to claim diversity without providing any support for it. Also, the paper does not have any mention of ethical concerns or societal impact. Given that such a dataset was created, especially using personal VLOG style videos, this seems like it should have some discussion.
>
> We mention the dataset is `diverse` as it is collected from less constrained video sources, in comparison to existing datasets mostly built on videos associated with a fixed set of video categories. For example, YouTubeHighlights [32] are based on videos from 6 categories (e.g., `surfing`), ActivityThumb [37] and ANetCaptions [14] are collected based on 200 human action classes. In contrast, our videos are from general domains, i.e., ‘daily vlog’. It is shown in [Fouhey et al.] that this collection procedure leads to greater diversity in terms of actions. In Table 1, we also present some analysis on unique verbs and nouns in the dataset, which shows the videos used in the dataset are very different across domains.
>
> Fouhey et al. From lifestyle vlogs to everyday interactions.CVPR  2018.
>
>
>
>
> ---------------
>
> > (2) The dataset is also fairly small by current standards, 10,000 videos is not many.
>
> We do not aim to provide a very large amount of data due to the high cost of human annotation, (e.g., in supplementary Section B, annotating 10K videos already costs around $16,000, which is a large amount for an academic budget). Meanwhile, as we have seen recent trends [citations 22, 21, 17] on utilizing large-scale weakly aligned web videos for pre-training and then fine-tuning on human-annotated data, we aim to provide it as a well-annotated dataset mainly for evaluation purposes. We also noticed this dataset itself might not be enough to train large or complex models, thus we utilized weakly supervised pre-training via ASR captions, as stated in Section 4.3, L242-248.
>
>
>
> ---------------
>
> > (3) The proposed baseline approach is fairly straightforward adaptation of existing approaches, like DETR to videos. There is some value in seeing it work and providing a good baseline for the new dataset. It would also be good to compare the proposed baseline to previous works on other, existing datasets to help make comparisons.
>
> To our best knowledge, we are the first to use Moment-DETR in the video domain. It is worth noting that Moment-DETR does not come free to the video domain. There are a few key challenges to make it work, e.g., adding a loss (saliency loss) that explicitly encourage the learning of similarity between clips and queries, in Table 4, we show saliency loss is critical to good performance, note that this is also true when we do not use the saliency annotation -- by removing the saliency annotation term (t_low/t_high) from the saliency, we see adding this loss still significantly improve the model performance:
>
> | saliency loss type | Moment Retrieval |         | Highlight Detection |       |
> |--------------------|------------------|---------|---------------------|-------|
> |                    | R1 @0.5           | mAP avg | mAP                 | HIT@1 |
> | None               | 44.84            | 25.05   | 17.84               | 20.19 |
> | t_out/t_in only        | 52.9             | 31.46   | 35.62               | 52.58 |
>
>
>
> To better compare Moment-DETR with existing methods, we also tested Moment-DETR on CharadesSTA [cited in paper as 6], test split:
>
> | Method                              | R1 @0.5 | R1 @0.7 |
> |-------------------------------------|--------|--------|
> | CAL [Escorcia et al. arXiv 2019]    | 44.9   | 24.37  |
> | 2D TAN [Zhang et al. AAAI 2020]     | 39.7   | 23.31  |
> | CBP [Wang et al. AAAI 2020]         | 36.8   | 18.87  |
> | DRN [Zeng, et al. CVPR 2020]        | 45.4   | 26.4   |
> | VSLNet [Zhang et al. ACL 2020]      | 47.31  | 30.19  |
> | IVG-DCL [Nan et al. CVPR 2021]      | 50.24  | 32.88  |
> | Moment-DETR                         | 52.12  | 29.84  |
> | Moment-DETR w/ HT100M 10K videos PT | 54.06  | 31.32  |
>
>
> When trained from scratch, Moment-DETR is competitive compared to the baselines, with the best R1 @0.5 (+~2% compared to IVG-DCL [Nan et al. CVPR 2021], which appears on arXiv after the NeurIPS deadline). Note that this is our initial trial during the tight response period, we expect better performance when Moment-DETR is properly tuned. When pretrained with a small scale of 10K videos (this pretraining is efficient, only ~8 hours with a single RTX 2080Ti), Moment-DETR’s performance is further improved. Besides, we also notice Moment-DETR’s performance on R1 @0.7 is not as good as R1 @0.5, suggesting it may be less accurate in localizing the exact temporal boundaries, we see similar observations in QVHilights and is mentioned as a limitation of the work that needs future investigations (L284-291).
>
>
> Escorcia et al. Temporal localization of moments in video collections with natural language. arXiv 2019.
>
> Nan et al., Interventional Video Grounding with Dual Contrastive Learning. CVPR 2021.
>
> Zhang et al., Span-based localizing network for natural language video localization. ACL 2020.
>
> Zhang et al., Learning 2d temporal adjacent networks for moment localization with natural language. AAAI 2020.
>
> Wang et al., Temporally grounding language queries in videos by contextual boundary-aware prediction. AAAI 2020.
>
> Zeng, et al. Dense regression network for video grounding. CVPR 2020.
>
>
>
>
> ---------------
>
> > In the checklist: "Did you include the estimated hourly wage paid to participants and the total amount spent on participant compensation? [Yes] See Section 3.1."
> I do not see any estimated payments, wage, or time spent on the task in Section 3.1 or the paper or supplementary materials.
>
>
> Due to space limit, we moved the annotation cost and time details to Supplementary file Section B `Annotation Cost`. Sorry for the confusion and we will add it back to the main paper given the extra page in the final version.
>
>
>
>
> ---------------
>
> > There is some discussion on the limitations of the proposed pretraining, but otherwise there is no discussion on the limitations of the approach or dataset and no meaningful discussions on the societal impacts of the dataset.
>
> We discussed two limitations on our approach: (1) L242-244: requiring larger-scale data for training; (2) L284-292: Moment-DETR performs quite well on low IoU threshold, but its performance on higher IoU threshold is relatively lower, e.g., compared to XML+, Moment-DETR shows a clear advantage on R1 @0.5, but only achieves a similar performance on R1 @0.7. We will make this clearer.
>
> For societal impacts, we believe the tools developed using our dataset might have an educational impact, where teachers as well as students can study and analyze long videos of entire weeks/semesters to find specific query-related salient parts of the classroom interactions (to improve the teaching quality for teachers, and to re-study later after class for students or for students who missed classes). We’ll add discussions in revision.
>
>
> ---------------
>
> > The paper does raise a few potential ethical concerns, regarding the dataset. The dataset consists of "over 10,000 YouTube videos, covering a wide range of topics, from everyday activities and travel in lifestyle vlog videos to social and political activities in news videos"
> This is concerning because it can provide annotations that could be used to violate privacy (e.g., using personal vlog videos) or enabling other agencies to use this data to build models to understand vlog-style videos for undesirable purposes. Also having societal and political activities (e.g., protests as mentioned in the paper) in the data could be used for undesirable surveillance purposes. Given that there are protests, there could also be violent imagery in this dataset (not mentioned if this has been removed or not).
>
>
> We only use public videos from YouTube. To use these videos, we rely on Fair Use (https://www.copyright.gov/fair-use/), as indicated in supplementary file L22. We also plan to restrict the use of the data for research purposes only by having users sign an agreement. Overall, we aim to collect a dataset that is general enough to cover a wide range of activities, thus we collected a set of videos out of several video domains, including popular news topics, such as weather and some social activities. During this process, we also explicitly removed videos that contain improper content, such as these containing violent imagery.

---

### Official Review · Reviewer_zZwJ · 2021-07-16

**Rating:** 5
**Confidence:** 4

**Summary:**

This paper collects the QVHIGHLIGHTS, a unified benchmark dataset for text-based moment retrieval and highlight detection in videos. It contains over 1000 Youtube videos with a) human-written free-form queries and their relevant moments, and b) five-point scale saliency scores for all query-relevant clips. Different from existing text-based moment retrieval datasets, QVHIGHLIGHTS has a smaller temporal bias on moments and allows a text query to match multiple disjoint moments. Compared to other highlight detection datasets, it presents temporally dense annotations dependent on queries.
As a baseline, they propose a Moment-DETR model which can do the two tasks jointly.

**Limitations And Societal Impact:**

Yes

**Main Review:**

Strength:
1. The dataset unifies two tasks based on text queries -- moment retrieval and highlight detection, and show how joint training can affect the performance of the baseline model on these two tasks respectively.
2.The collected dataset overcomes some of the limitations in other moment retrieval datasets by having less temporal bias and allowing a text query to match multiple disjoint moments.
3. The dataset accounts for the subjectiveness in highlight detection and provide dense annotations.
4. Good quality control through the qualification test and the verification between multiple workers.


Weakness:
1. Line 259-260 'we define a prediction to be positive if it has a high IoU with
one of the ground truth moments'.  It is not very clear here whether 'one of the ground truth moments' means one from 3 annotators or one from disjoint moments.
2. For highlight detection: Line 262-263, 'we have ground-truth saliency scores from 3 users, we evaluate performance against each then take the average'. Does it mean the model's performance is dependent on the inter-user agreement?
3. Missing baseline experiments/ablations:
 - The results from using only CLIP and Slow-Fast networks as visual encoder, as in how important the temporal information is in the two tasks.
 - Performance on moment retrieval after the model is pre-trained via ASR, to see how much improvement has supervised training brought.
 - Evaluation of SoTA models on moment retrieval which have official implementations online: 2D-TAN[1], DRN[2], CBP[3]
4. Performance of the baseline model:
 - Since one advantage of the dataset is having queries matching disjoint moments, how well can the baseline model spotting all the disjoint moments given a query, is there a metric to evaluate it?
 - Table 1 shows the dataset has a wide range of granularity in the verbs (e.g, cut and report) and nouns(dog and weather), does the performance of the baseline model varies by the granularity?
5. While the highlights are labelled on a five-point scale, is that only the 'very good' ones are used in training and evaluation?
6. Related work: [4] uses a DETR-like architecture for temporal detection, and treat the task as a set-prediction task.

[1] Learning 2D Temporal Localization Networks for Moment Localization with Natural Language. Zhang et al. AAAI 2020.
[2] Local-Global Video-Text Interactions for Temporal Grounding. Mun et al. CVPR 2020.
[3] Temporally Grounding Language Queries in Videos by Contextual Boundary-aware Prediction. Wang et al. AAAI 2020
[4] Activity Graph Transformer for Temporal Action Localization. Nawhal et al. CVPR 2021.

Summary:

The QVHIGHLIGHTS dataset addresses some limitations in current moment retrieval/highlight detection datasets. It is presented in a clear way and will be beneficial to future research in these areas. However, some detailed analysis on the dataset is missing, and more baseline experiments are needed to verify the design choice in the baseline architecture.


**Time Spent Reviewing:**

6 hours

---

> ### Author Response · Authors · 2021-08-10
> **Author Response (zZwJ)**
>
> Thanks for reviewing our work and giving constructive feedback. We answer your questions below:
>
> ---------------
>
> > Line 259-260 'we define a prediction to be positive if it has a high IoU with one of the ground truth moments'. It is not very clear here whether 'one of the ground truth moments' means one from 3 annotators or one from disjoint moments.
>
> In our final dataset, for each video, we only have a single set of moment annotations. Thus 'one of the ground truth moments' means one from disjoint moments. Also note that we do have 3 sets (from different annotators) of moments for 600 query-video pairs, but these are only used for verifying that our annotations are of high user agreement (L125) and are not intended to be used anywhere else.
>
> ---------------
>
> > For highlight detection: Line 262-263, 'we have ground-truth saliency scores from 3 users, we evaluate performance against each then take the average'. Does it mean the model's performance is dependent on the inter-user agreement?
>
> For HIT@1, we define a prediction to be correct if it matches with one of the ground-truth annotations among the three. This metric is useful when considering the subjectiveness of highlights for different users and it only evaluates the highest scored prediction.
>
> For mAP, we first evaluate each annotation and take the average. This metric considers more about how the models perform globally. Since it takes the average of results for the three annotations, it also considers the inter-user agreement, as you mentioned.
>
> In conclusion, the two metrics focus on different aspects of and using both of them allows for a more comprehensive evaluation. Sorry for the potentially confusing description and we will add details in revision.
>
>
>
> ---------------
>
> > The results from using only CLIP and Slow-Fast networks as visual encoder, as in how important the temporal information is in the two tasks.
>
> |                    | Moment Retrieval |         | Highlight Detection |       |
> |--------------------|------------------|---------|---------------------|-------|
> | video feature type | R1 @0.5           | mAP avg | mAP                 | HIT@1 |
> | CLIP only          | 53.23            | 30.58   | 35.51               | 55.87 |
> | SlowFast only      | 52.06            | 30.54   | 35.85               | 56.71 |
> | CLIP+SlowFast      | 53.94            | 32.2    | 35.65               | 55.55 |
>
> Overall, using CLIP or SlowFast alone shows similar performance on moment retrieval, and is slightly worse than using both of them, suggesting they are complementary for this task. For highlight detection, using SlowFast feature performs better, but the difference is small.
>
>
> ---------------
>
> > Performance on moment retrieval after the model is pre-trained via ASR, to see how much improvement has supervised training brought.
>
> The pretraining only results on QVHighlights test split:
>
> |                        | Moment Retrieval |         | Highlight Detection |       |
> |------------------------|------------------|---------|---------------------|-------|
> | method                 | R1 @0.5           | mAP avg | mAP                 | HIT@1 |
> | pretraining only       | 3.44             | 0.7     | 26.53               | 32.23 |
> | finetuning only        | 52.89            | 30.73   | 35.69               | 55.6  |
> | pretraining+finetuning | 59.78            | 36.14   | 37.43               | 60.17 |
>
> We notice supervised training/finetuning (row 3 vs. row 1) brought a huge gain to both tasks. This is expected as (1) the languages in annotated queries (declarative sentences referring to specific actions/objects, etc.) are very different from the free-form monologues or dialogues in the ASR; (2) The ASR text may be only weakly related to the videos. Meanwhile, we see the pretraining only model achieves quite low performance on moment retrieval but reasonably good results on highlight detection. This may indicate, though it failed to precisely locate the queries, the model learned to align the queries and the video clips that share similar meanings.
>
>
>
> ---------------
>
> > Evaluation of SoTA models on moment retrieval which have official implementations online: 2D-TAN[1], DRN[2], CBP[3]
>
> Thanks for your suggestion. We ran 2D-TAN model on QVHighlights dataset, the test split results are shown below
>
> | method            | text feature | visual feature | R1 @0.5 | mAP avg |
> |-------------------|--------------|----------------|--------|---------|
> | 2D TAN            | GloVe        | SlowFast+CLIP  | 32.81  | 22.46   |
> | 2D TAN            | CLIP         | SlowFast+CLIP  | 35.86  | 27.22   |
> | Moment-DETR       | CLIP         | SlowFast+CLIP  | 52.89  | 30.73   |
> | Moment-DETR w/ PT | CLIP         | SlowFast+CLIP  | 59.78  | 36.14   |
>
> Under a fair comparison, using the same features, we notice Moment-DETR (row3) shows a clear advantage over 2D-TAN (row2). We’ll add more results in revision.
>
>
> ---------------
>
> > Since one advantage of the dataset is having queries matching disjoint moments, how well can the baseline model spotting all the disjoint moments given a query, is there a metric to evaluate it?
>
> Yes! We follow [2] to use mAP for evaluating predictions, this metric takes into account multiple moments during evaluation. This is also similar to how object detection is evaluated (one image may contain multiple objects of interest).
>
>
> ---------------
>
> >  Table 1 shows the dataset has a wide range of granularity in the verbs (e.g, cut and report) and nouns(dog and weather), does the performance of the baseline model varies by the granularity?
>
> Yes, the performance on different video domains vary. In the table below, we categorized the test split performance (Table 3) by grouping queries into one of the 3 domains, the percentage of queries in each domain are shown in the brackets. We show results for Moment Retrieval (MR) and Highlight Detection (HD) for all the baselines. Overall, we notice the performance on `daily` and `travel` domain are higher than `news`. This may be from (1) different granularity as you suggested; (2) data size in the domains.
>
> |                   | MR R1 @0.5  |                |              | HD HIT@1      |                |              |
> |-------------------|----------------|----------------|--------------|---------------|----------------|--------------|
> | Method            | daily (46.5%)  | travel (43.1%) | news (10.4%) | daily (46.5%) | travel (43.1%) | news (10.4%) |
> | BeautyThumb       | -              | -              | -            | 24.13         | 17.44          | 20.62        |
> | DVSE              | -              | -              | -            | 21.90         | 21.50          | 22.50        |
> | MCN               | 8.23           | 14.44          | 13.12        | -             | -              | -            |
> | CAL               | 24.83          | 26.92          | 22.50        | -             | -              | -            |
> | XML               | 45.05          | 40.45          | 33.12        | 58.58         | 53.08          | 49.38        |
> | XML+              | 49.37          | 46.62          | 35.00        | 57.18         | 54.44          | 48.12        |
> | Moment-DETR       | 51.80          | 56.57          | 42.50        | 56.15         | 56.93          | 47.62        |
> | Moment-DETR w/ PT | 63.22          | 59.08          | 48.63        | 60.27         | 61.95          | 51.75        |
>
>
>
> ---------------
>
> > While the highlights are labelled on a five-point scale, is that only the 'very good' ones are used in training and evaluation?
>
> For training, we use all scores. In L234, we mentioned we construct a pair with a high score clip and a low score clip. For evaluation, in the paper we only reported performance with `>= Very Good`, but we do have results for `>= Good`, and `>= Fair`, etc. and our evaluation script (in supplementary file) supports evaluating scores at all scales. We hope this could be useful for future work for more detailed analysis. Due to space limits, we only report the most strict one, i.e., `>=Very Good`.
>
>
> > Related work: [4] uses a DETR-like architecture for temporal detection, and treat the task as a set-prediction task.
>
> Thanks for mentioning this related work. We noticed [4] uses similar DETR-like architectures detecting a fixed set of actions and their corresponding temporal locations, and in this work we focus on detecting the temporal locations of unconstrained language queries from the videos. [4] works on pure video, while this work requires learning from both video and language. We will cite it and add more discussions in revision.

---

> > ### Comment · Reviewer_zZwJ · 2021-08-22
> > **Discussion**
> >
> > I'd like to thank the authors for their efforts in running asked experiments and clarifying the confusion.
> >
> > The response has addressed most of my concerns, the remaining ones are:
> >
> > 1) **Bias in the dataset**  Overall this is a very good dataset. However, experiments show that adding visual features from the spatial-temporal network (Slow-Fast) only gives 0.7% improvement over the spatial network (CLIP), it may indicate that fine-grained temporal information helps very little, moment retrieval on the proposed dataset has a strong bias towards spatial information.
> >
> > 2) **Performance on other Benchmarks** Performance of Moment-DETR on CharadesSTA shows that the model has a relatively lower recall @IOU=0.7, thus has a disadvantage in localizing the moment with high accuracy.  As the authors mentioned "we expect better performance when Moment-DETR is properly tuned. ", if there is any number to be updated before the discussion period ends, please feel free to do so.

---

> > > ### Author Response · Authors · 2021-08-31
> > > **Response to New Discussion (zZwJ)**
> > >
> > > Thank you for your time for making this detailed feedback. We are happy that our response has addressed most of your concerns. For your remaining questions:
> > >
> > > **Bias in the dataset**: We believe the small improvement from adding Slow-Fast features may also be because of the already very impressive representation power of CLIP (Radford et al. 2021), even for videos. In the table below, we show CLIP’s linear probe performance using a single frame for video action recognition tasks. We notice that CLIP’s performance is better than the two spatio-temporal models, I3D and S3D on the two standard action recognition datasets, even though the S3D model is petrained on a large video corpus (100M video-text pairs from HT100M). Thus, the small performance gain from adding Slow-Fast may not strongly support the claim that “fine-grained temporal information” is not important. Meanwhile, also note that our goal is to cover general and natural queries from users, we did not explicitly control what should be queried -- thus the dataset should reflect what general users care about when they perform a search. We do agree that it is important that future work explores designing datasets explicitly for understanding “fine-grained temporal information”, we’ll add a discussion in our final version.
> > >
> > >
> > > | Method                                                       | UCF101 | Kinetics700 |
> > > |--------------------------------------------------------------|--------|-------------|
> > > | I3D (Carreira et al, 2019)                                 | -      | 70.2        |
> > > | S3D (HT100M pretrained + e2e finetuned, (Miech et al 2020)) | 91.3   | -           |
> > > | CLIP (single frame, linear probe, (Radford et al 2021))                            | 92     | 73          |
> > >
> > > Carreira et al. A short note on the kinetics-700 human action dataset. arXiv 2019
> > >
> > > Miech et al. End-to-end learning of visual representations from uncurated instructional videos. CVPR 2020
> > >
> > > Radford et al. Learning transferable visual models from natural language supervision. arXiv 2021
> > >
> > > UCF101: Soomro et al. UCF101: A dataset of 101 human actions classes from videos in the wild. arXiv 2012
> > >
> > > Kinetics700: Carreira et al. A short note on the kinetics-700 human action dataset. arXiv 2019
> > >
> > >
> > > ------------------------
> > >
> > >
> > > **Performance on other Benchmarks**: We have updated our results on Charades-STA, please see the table below. The two rows with the suffix `(updated)` are the results from our recent experiments with a bit hyper-parameter tuning (see details in the next paragraph). We notice that the updated Moment-DETR with a small-scale pre-training (last row) shows a notable improvement over the SotA method IVG-DCL on both R1@0.5 and R1@0.7; even without pretraining, Moment-DETR (updated) still show a very competitive performance. Since this is the first attempt to adopt a DETR-like model for video moment retrieval, we provide a strong starting point and hope future work could further improve it.
> > >
> > > *Tuning details*:
> > > - we increased the loss weight (i.e., $\lambda_{saliency}$ in `Eq (4)`) for the saliency loss (i.e., $\mathcal{L}_{saliency}$ in `Eq (3)`) from `1` to `4`, to account for the fact that it has a smaller saliency loss value compared to that on QVHilights datasets -- since no saliency annotation is available for Charades-STA, this loss is only calculated using the 2nd term in `Eq (3)`.
> > > - we also decayed the learning rate by a factor of 10 every 40 epochs, which we previously kept the same during training.
> > >
> > >
> > > | Method                                        | R1@0.5 | R1@0.7 |
> > > |-----------------------------------------------|--------|--------|
> > > | CAL [Escorcia et al. arXiv 2019]              | 44.9   | 24.37  |
> > > | 2D TAN [Zhang et al. AAAI 2020]               | 39.7   | 23.31  |
> > > | CBP [Wang et al. AAAI 2020]                   | 36.8   | 18.87  |
> > > | DRN [Zeng, et al. CVPR 2020]                  | 45.4   | 26.4   |
> > > | VSLNet [Zhang et al. ACL 2020]                | 47.31  | 30.19  |
> > > | IVG-DCL [Nan et al. CVPR 2021]                | 50.24  | 32.88  |
> > > | Moment-DETR                                   | 52.12  | 29.84  |
> > > | Moment-DETR `(updated) `                        | 53.63  | 31.37  |
> > > | Moment-DETR w/ HT100M 10K videos PT           | 54.06  | 31.32  |
> > > | Moment-DETR w/ HT100M 10K videos PT `(updated)` | **55.65**  | **34.17**  |
> > >
> > > ------------------------
> > >
> > > Given that we have addressed most of your concerns from your original review, and this new response to your recent questions, we hope there can be updates/increase on your original judgment/score of our work. We really appreciate it! We believe the proposed dataset and method would be useful for many people in the community.

---

### Official Review · Reviewer_uoxL · 2021-07-19

**Rating:** 7
**Confidence:** 5

**Summary:**

This paper introduces a new dataset for video moment retrieval and highlight detection named QVHighlights. The dataset contains around 10K YouTube videos. Each video is annotated with a free form text query and (i) the time intervals in the video corresponding to the query (multiple intervals can match the query) and (ii) for each annotated time interval a *saliency* scores (from 1 (Very bad) to 5 (very good)) describing how relevant are the individual frames within an annotated clip are to the query.

In addition to the dataset, the authors introduce a method named MomentDETR inspired from DETR in order to perform moment retrieval in their introduced dataset. The method performs better than existing techniques.

**Ethical Concerns:**

No ethical concerns apart from the datasheet request above.

**Limitations And Societal Impact:**

The authors have not commented on the limitations and societal impact of their work.

Since the authors introduce a new dataset, I would recommend the authors to fill a datasheet for it (see [Datasheet for dataset, Gebru et al.](https://arxiv.org/abs/1803.09010))

**Main Review:**

## Strengths

- The paper is clearly written
- The dataset is going to be useful to the community as it seems it is quite high quality (based on the description and the rater agreement).
- I believe that the introduced method MomentDETR is a good extension to the DETR approach for moment retrieval in videos.
- The pre-training strategy on weakly supervised data with ASR is interesting and leads to a substantial improvement.


## Weaknesses

Overall the paper is good but there a few points that could improve it further:

- **Weakly supervised pretraining**: The weakly supervised pretraining seems to bring a substantial gain in performance (Table 3). However from what I understood, only a small dataset is used for this pretraining (only 5.4K videos). Have you thought about using larger datasets for this pretraining? How would that impact the performance of the method? One could for example use the HowTo100M dataset for that [22]. I believe that a scale study would be very valuable there even if not scaling to the full HowTo100M dataset (e.g. providing the performance with only 2.5K videos and 10K videos to see the impact of increasing the size of the pretraining). The VLOGs dataset [a] could also be used for that purpose.

- **MomentDETR evaluation**: MomentDETR is an interesting method in itself. For that reason, I think it would be worthwhile to run MomentDETR on other moment retrieval benchmarks to show it is competitive on known benchmarks. I think one of the main issue of the paper is that it introduces a new dataset and a new method so it is hard to convince the reader that the method is indeed better than pas work such as [30,19,12,4,16] as we cannot compare direclty to some numbers provided in these papers. I do trust the authors for having run these baselines properly but I think that running the method on other datasets would increase the strength of the paper.

- **Precision about the saliency loss**:
  - How are $t_{high}$ and $t_{low}$ defined based on the scores from 1 to 5?
  - How important is the term using low/high versus the term using out/in. Could you provide an ablation about this in Table 4 (keeping all other elements i.e. L1/gIoU and CLS)? If the term low/high is indeed important that would justify more the need for annotating individual frames which is one of the specificity of the QVHighLights dataset.

- **Highlight and saliency score**: Some frames within positive annotated segments are annotated as `Very bad` (and `Bad`) for the given query. This seems to contradict that these frames are indeed positive for that query. In that case why not removing these frames all together from the annotation and create more moments? It would also be useful to provide the score distribution (% of frames that receive a score of 1,2,3,4,5...).

- Minor typos:
  - L79: lsmiting -> limiting
  - L100: sailiency -> saliency
  - L326: detetcion -> detection

- Missing related work:

  - [a] From lifestyle vlogs to everyday interactions. Fouhey et al. CVPR18

## Overall assessment

Overall the paper makes good contributions. However I believe that these contributions could be strengthen following the advices provided in the weaknesses section. For now I lean towards acceptance and I am ready to increase my score if some/all concerns are addressed during the rebuttal.


## Post rebuttal

After reading other reviews and the author's response I am happy to support acceptance of this paper. I really encourage the authors to include all the new ablations and set of experiments requested by the reviewers as they will make for a stronger paper, which in turn will be more influential in the community.

**Time Spent Reviewing:**

3.5

---

> ### Author Response · Authors · 2021-08-10
> **Author Response (uoxL)**
>
> Thanks for reviewing our work and giving constructive feedback. We answer your questions below:
>
>
>
> ------
>
> > Weakly supervised pretraining: The weakly supervised pretraining seems to bring a substantial gain in performance (Table 3). However from what I understood, only a small dataset is used for this pretraining (only 5.4K videos). Have you thought about using larger datasets for this pretraining? How would that impact the performance of the method? One could for example use the HowTo100M dataset for that [22]. I believe that a scale study would be very valuable there even if not scaling to the full HowTo100M dataset (e.g. providing the performance with only 2.5K videos and 10K videos to see the impact of increasing the size of the pretraining). The VLOGs dataset [a] could also be used for that purpose.
>
> Thanks! We followed your suggestion and performed the pretraining experiments using 2.5K and 10K HowTo100M videos. We show their finetuning performance on QVHilights val split:
>
> |                    | Moment Retrieval |         | Highlight Detection |       |
> |--------------------|------------------|---------|---------------------|-------|
> | pretraining videos | R1 @0.5           | mAP avg | mAP                 | HIT@1 |
> | None               | 53.94            | 32.2    | 35.65               | 55.55 |
> | 2.5K HT100M        | 54.58            | 32.82   | 36.27               | 56.45 |
> | 10k HT100M         | 56.06            | 34.16   | 35.97               | 56.84 |
> | 5.4K QVHighlights  | 59.68            | 36.3    | 37.7                | 60.32 |
>
> We notice that adding HowTo100M videos for pretraining (row 2,3) improves performance on both moment retrieval and highlight detection. But the improvement is smaller for the highlight detection task compared to that of the moment retrieval task. Meanwhile, when increasing 2.5K videos to 10K videos, we see a positive trend on moment retrieval, but the highlight detection performance stays similar. Potentially due to the size of HowTo100M videos we use are still relatively small. We expect a larger pre-training corpus to significantly improve these results.
> Another observation is that, when pretraining on same-domain videos, i.e, QVHighlights videos, a small 5.4K videos shows significantly better results than using 10K out-of-domain HowTo100M videos. This suggests the importance of pretraining on same or closer domain data.
>
> ----------------
>
> > MomentDETR evaluation: MomentDETR is an interesting method in itself. For that reason, I think it would be worthwhile to run MomentDETR on other moment retrieval benchmarks to show it is competitive on known benchmarks. I think one of the main issue of the paper is that it introduces a new dataset and a new method so it is hard to convince the reader that the method is indeed better than pas work such as [30,19,12,4,16] as we cannot compare directly to some numbers provided in these papers. I do trust the authors for having run these baselines properly but I think that running the method on other datasets would increase the strength of the paper.
>
> To address this issue, we tested Moment-DETR on CharadesSTA (cited in paper as [6]). Results are shown below:
>
> | Method                              | R1 @0.5 | R1 @0.7 |
> |-------------------------------------|--------|--------|
> | CAL [Escorcia et al. arXiv 2019]    | 44.9   | 24.37  |
> | 2D TAN [Zhang et al. AAAI 2020]     | 39.7   | 23.31  |
> | CBP [Wang et al. AAAI 2020]         | 36.8   | 18.87  |
> | DRN [Zeng, et al. CVPR 2020]        | 45.4   | 26.4   |
> | VSLNet [Zhang et al. ACL 2020]      | 47.31  | 30.19  |
> | IVG-DCL [Nan et al. CVPR 2021]      | 50.24  | 32.88  |
> | Moment-DETR                         | 52.12  | 29.84  |
> | Moment-DETR w/ HT100M 10K videos PT | 54.06  | 31.32  |
>
> When trained from scratch, Moment-DETR is competitive compared to the baselines, with the best R1 @0.5 (+~2% compared to IVG-DCL [Nan et al. CVPR 2021], which appears on arXiv after the NeurIPS deadline). Note that this is our initial trial during the tight response period, we expect better performance when Moment-DETR is properly tuned. When pretrained with a small scale of 10K videos (this pretraining is efficient, only ~8 hours with a single RTX 2080Ti), Moment-DETR’s performance is further improved. Besides, we also notice Moment-DETR’s performance on R1 @0.7 is not as good as R1 @0.5, suggesting it may be less accurate in localizing the exact temporal boundaries, we see similar observations in QVHilights and is mentioned as a limitation of the work that needs future investigations (L284-291).
>
> Escorcia et al. Temporal localization of moments in video collections with natural language. arXiv 2019.
>
> Nan et al., Interventional Video Grounding with Dual Contrastive Learning. CVPR 2021.
>
> Zhang et al., Span-based localizing network for natural language video localization. ACL 2020.
>
> Zhang et al., Learning 2d temporal adjacent networks for moment localization with natural language. AAAI 2020.
>
> Wang et al., Temporally grounding language queries in videos by contextual boundary-aware prediction. AAAI 2020.
>
> Zeng, et al. Dense regression network for video grounding. CVPR 2020.
>
>
> ----------------
>
> > How are t_high and t_low defined based on the scores from 1 to 5?
>
> We sum up the highlight scores from all 3 annotators for each relevant clip, then set t_high to be the clip with the maximum score, and t_low to be the clip with the minimum score. We’ll add these details in revision.
>
>
> ----------------
>
> > How important is the term using low/high versus the term using out/in. Could you provide an ablation about this in Table 4 (keeping all other elements i.e. L1/gIoU and CLS)? If the term low/high is indeed important that would justify more the need for annotating individual frames which is one of the specificity of the QVHighLights dataset.
>
> Following your suggestion, we did another set of ablation studies. The results are:
>
> | saliency loss type | Moment Retrieval |         | Highlight Detection |       |
> |--------------------|------------------|---------|---------------------|-------|
> |                    | R1 @0.5           | mAP avg | mAP                 | HIT@1 |
> | None               | 44.84            | 25.05   | 17.84               | 20.19 |
> | out/in only        | 52.9             | 31.46   | 35.62               | 52.58 |
> | low/high only      | 52.52            | 30.35   | 29.32               | 40.77 |
> | low/high + out/in  | 53.94            | 32.2    | 35.65               | 55.55 |
>
> Our observations are (1) row2,3 vs. row1: adding either one of the terms improves performance on both tasks; (2) row4 vs. row2,3: adding both terms leads to the best overall performance; (3)row4 vs. row2: adding low/high term on top of out/in improves performance on highlight detection HIT@1 by ~3% absolute, while mAP stays similar. Meanwhile, adding it also gives slightly better results on moment retrieval. These observations show the usefulness of the term low/high. Also note that, as a baseline, we only explored a simple strategy to use the highlightness annotations (see the answer to the last item), we expect further improvement if more advanced approaches are used.
>
>
> ----------------
>
> > Highlight and saliency score: Some frames within positive annotated segments are annotated as Very bad (and Bad) for the given query. This seems to contradict that these frames are indeed positive for that query. In that case why not removing these frames all together from the annotation and create more moments? It would also be useful to provide the score distribution (% of frames that receive a score of 1,2,3,4,5...).
>
> During moment annotation, we performed a verification experiment (see L124-129), we found the annotated segments/moments have a very high inter-user agreement, which suggests a high quality of these moments. After we collected the saliency scores, we did another analysis, i.e, the score distributions, We noticed 94.41% of the annotated clips are rated by two or more users as `fair or better` (>=3, meaning they may be less saliency, but still relevant, see supplementary file Fig 6). Only 0.96% of the clips have two or more users rated as `very bad`. This result is consistent with our earlier moment verification experiments. Since only a tiny percentage of the annotated segments/moments are truly irrelevant, they do not hinder the use of the data. Thus we choose to keep and release the data in its current form.
>
> Score distribution:
> (since we have scores from 3 users for each clip, we show the score percentage as two or more users agree, e.g., for 5.59% of the clips, two or more users agree they should be rated as equal or lower than `bad`.)
>
> | score range     | %     |
> |-----------------|-------|
> | <=1 (very bad)  | 0.96  |
> | <=2 (bad)       | 5.59  |
> | <=3 (fair)      | 23.44 |
> | <=4 (good)      | 62.1  |
> | <=5 (very good) | 100   |
>
>
> ----------------
>
> > Minor typos:
>
> Thanks! We will fix these typos.
>
>
> ----------------
>
> > Missing related work: [a] From lifestyle vlogs to everyday interactions. Fouhey et al. CVPR18
>
> Thanks! We will add this citation.

---

### Official Review · Reviewer_sT4H · 2021-07-26

**Rating:** 6
**Confidence:** 4

**Summary:**

In this paper, the authors focus on moment retrieval and highlight detection tasks from videos given text queries. They present the QVHIGHLIGHTS dataset which includes 10,000 videos collected from Youtube. Each video is annotated with a query, time span of a moment, and saliency scores.
In addition, the authors propose an Encode-Decoder Transformer model (Moment-DETR) that predicts moments and scores in an end-to-end manner.
The Moment-DETR model provides superior performance in comparison to previous methods on the QVHIGHLIGHTS dataset for moment temporal localization and highlight detection tasks.

**Ethical Concerns:**

There are no ethical issues with this paper.

**Limitations And Societal Impact:**

The authors described the technical limitations but I couldn't find any information regarding the negative societal impact of their work.

**Main Review:**

- line 114: During data collection, how do you segment raw videos into short clips? Is it random or based on specific criteria?
- In table 4, can you explain why removing L1 or gIoU losses decrease the highlight detection performance?
- Can TVR and ANetCaptions datasets help in pre-training the MomentDetr model? (removing HD related losses)
- Although the Moment-DETR model performs well on QVHIGHLIGHTS dataset, there is no evidence that it will generalize to other datasets. For instance, how will it perform on the TVR dataset in comparison to SOTA?
- Authors extract video and text features using the CLIP model. Is there a way to use the nearest neighbor approach to evaluate moment retrieval (distance between text feature and clip features)? how well does it perform in comparison to Moment-DETR?


Typos:
- line 79: lsimiting -> limiting

**Time Spent Reviewing:**

2

---

> ### Author Response · Authors · 2021-08-10
> **Author Response (sT4H)**
>
> Thanks for reviewing our work and giving constructive feedback. We answer your questions below:
>
>
>
> ------
>
> > line 114: During data collection, how do you segment raw videos into short clips? Is it random or based on specific criteria?
>
> We segment the raw videos sequentially. Besides, we also follow [Lei et al.] to remove the first 60 seconds of the raw videos, as these contents are often subscription pleas, etc.
>
> Lei et al. What is more likely to happen next? video-and-language future event prediction. EMNLP 2020.
>
> ---------
>
> > In table 4, can you explain why removing L1 or gIoU losses decrease the highlight detection performance?
>
> They do not actually affect the highlight detection performance significantly. The results are indeed similar if we consider the variance across runs -- we performed 5 random runs of row 2-4, here are their highlight detection mAPs respectively, mean (std, standard deviation): 35.68 (0.47), 36.54 (0.26), 35.95 (0.48), we notice the difference between row 4 and row 2, row 4 and row 3  are not significant considering their standard deviation.
>
> ---------
>
> > Can TVR and ANetCaptions datasets help in pre-training the MomentDetr model? (removing HD related losses)
>
> Thanks for your suggestion! Due to time constraints, we used another dataset (10k videos from HowTo100M [Miech et al.]) for pretraining. We show its finetuning performance on QVHighlights val split:
>
> |                    | Moment Retrieval |         | Highlight Detection |       |
> |--------------------|------------------|---------|---------------------|-------|
> | pretraining videos | R1 @0.5           | mAP avg | mAP                 | HIT@1 |
> | None               | 53.94            | 32.2    | 35.65               | 55.55 |
> | 10k HT100M         | 56.06            | 34.16   | 35.97               | 56.84 |
>
> We notice that using 10k HowTo100M videos with ASR for pretraining improves performance on both moment retrieval and highlight detection. Since weakly labeled out-of-domain data helps in pretraining the model, we expect that using well-annotated datasets such as TVR, ANetCaptions will also give better results. We will add more results in revision.
>
> Miech et al. Howto100m: Learning a text-video embedding by watching hundred million narrated video clips. CVPR 2019.
>
> --------------
>
> > Although the Moment-DETR model performs well on QVHIGHLIGHTS dataset, there is no evidence that it will generalize to other datasets. For instance, how will it perform on the TVR dataset in comparison to SOTA?
>
>
> Here we experimented with another popular moment retrieval dataset, CharadesSTA (cited in paper as [6]) to demonstrate Moment-DETR’s generalization ability. Test split results are shown below:
>
> | Method                              | R1 @0.5 | R1 @0.7 |
> |-------------------------------------|--------|--------|
> | CAL [Escorcia et al. arXiv 2019]    | 44.9   | 24.37  |
> | 2D TAN [Zhang et al. AAAI 2020]     | 39.7   | 23.31  |
> | CBP [Wang et al. AAAI 2020]         | 36.8   | 18.87  |
> | DRN [Zeng, et al. CVPR 2020]        | 45.4   | 26.4   |
> | VSLNet [Zhang et al. ACL 2020]      | 47.31  | 30.19  |
> | IVG-DCL [Nan et al. CVPR 2021]      | 50.24  | 32.88  |
> | Moment-DETR                         | 52.12  | 29.84  |
> | Moment-DETR w/ HT100M 10K videos PT | 54.06  | 31.32  |
>
> When trained from scratch, Moment-DETR is competitive compared to the baselines, with the best R1 @0.5 (+~2% compared to IVG-DCL [Nan et al. CVPR 2021], which appears on arXiv after the NeurIPS deadline). Note that this is our initial trial during the tight response period, we expect better performance when Moment-DETR is properly tuned. When pretrained with a small scale of 10K videos (this pretraining is efficient, only ~8 hours with a single RTX 2080Ti), Moment-DETR’s performance is further improved. Besides, we also notice Moment-DETR’s performance on R1 @0.7 is not as good as R1 @0.5, suggesting it may be less accurate in localizing the exact temporal boundaries, we see similar observations in QVHighlights and is mentioned as a limitation of the work that needs future investigations (L284-291).
>
>
> Escorcia et al. Temporal localization of moments in video collections with natural language. arXiv 2019.
>
> Nan et al., Interventional Video Grounding with Dual Contrastive Learning. CVPR 2021.
>
> Zhang et al., Span-based localizing network for natural language video localization. ACL 2020.
>
> Zhang et al., Learning 2d temporal adjacent networks for moment localization with natural language. AAAI 2020.
>
> Wang et al., Temporally grounding language queries in videos by contextual boundary-aware prediction. AAAI 2020.
>
> Zeng, et al. Dense regression network for video grounding. CVPR 2020.
>
>
> --------------
>
> > Authors extract video and text features using the CLIP model. Is there a way to use the nearest neighbor approach to evaluate moment retrieval (distance between text feature and clip features)? how well does it perform in comparison to Moment-DETR?
>
> Thanks for your suggestion! We first compute the similarity scores using CLIP features between each sampled frame and the query, then use TAG [Zhao et al.] (backed by the classical watershed algorithm [Roerdink et al.]) to group top-scored frames into moments. Moment retrieval results on test split are shown below as `CLIP zero shot`:
>
> | Method            | R1 @0.5 | mAP avg |
> |-------------------|--------|---------|
> | CLIP zero shot    | 16.88  | 7.67    |
> | Moment-DETR       | 52.89  | 30.73   |
>
> As noted in this table, without training on the target dataset, CLIP’s results are reasonable, but still far below the performance of Moment-DETR.
>
> Zhao et al., Temporal action detection with structured segment networks. ICCV 2017
> Roerdink et al.: The watershed transform: Definitions, algorithms and parallelization strategies. Fundamenta informaticae 2000
>
> --------------
>
> > line 79: lsimiting -> limiting
>
> Thanks! We will fix the typo.
>
>
> --------------
>
> > The authors described the technical limitations but I couldn't find any information regarding the negative societal impact of their work.
>
> Societal impact:  In appendix Section C, we briefly mentioned that “the predictions reflect the distribution of the collected dataset. These predictions can be inaccurate, and hence users should not completely rely on our predictions for making real-world decisions (similar to previous work on modeling video-based predictions).” We will move it back to the main paper in revision.

---

### Review · Ethics_Reviewer_VN6J · 2021-08-10

**Recommendation:** 1. I agree with reviewers' recommenda…

**Ethical Issues:**

Yes

**Ethics Review:**

Generally, given the paper's task and contribution, there are two main ethical issues that arise: (1) the risk of making and facilitating inappropriate use of people's data, (2) the risk of facilitating development or extension of harmful forms of surveillance.

I commend the authors for what I believe is a thoughtful way of addressing these concerns (more below).

---

> ### Author Response · Authors · 2021-08-20
> **Author Response (Ethics Reviewer VN6J)**
>
> Thanks for reviewing our work and giving constructive feedback. We respond to your questions/suggestions below:
>
> > Dataset misuse: in their response the authors state that "We also plan to restrict the use of the data for research purposes only by having users sign an agreement". I believe this decision reduces the risk of misuse for surveillance purposes. In particular, while it is often expected that non-commercial licenses can prevent misuse, surveillance is often conducted by non-commercial entities (e.g. governments), and this risk is partially addressed by the authors proposed approach. It is worth noting, however, that portions of the design of surveillance technologies can be classified as "research", and authors should be mindful of this when granting permission to use the data.
>
> We are glad you agree with and appreciate our steps taken for preventing misuse. And yes, we will amend the agreement to explicitly restrict the use of the data and the resulting software for surveillance purposes or any other purpose that may violate human rights. Thanks for the suggestion.
>
> > I agree with reviewers' recommendation of filling out and providing a datasheet to accompany the data.
>
> Thanks! We will add the suggested datasheet to accompany the data. Meanwhile, we also copy our response to `reviewer uoxL` here: “Indeed, we already followed many best practices mentioned in Datasheet for dataset, Gebru et al., including details on collection process (Sec 3.1 and supplementary Section B), details on annotation files (supplementary material README.md) and license files, etc. We will cite and follow the recommended datasheet to further organize these contents.“
>
>
> > The risk of facilitating harmful forms of surveillance is not just associated to the dataset but also to the task. In their response to reviewers the authors say that they will include a discussion on societal impact by discussing the positive educational impact it may have. I also encourage the authors to reflect and acknowledge the possible misuses of the proposed methodology.
>
> Thanks for this suggestion. We do agree with the reviewers that both the dataset and task may be improperly used in scenarios such as surveillance (by the way, as you mentioned "I believe their choice to collect vlogs and news is thoughtful and reduces the risk", the videos we used, i.e., vlogs and news, may not be suitable for being used for surveillance purpose). We will add discussions with respect to this issue in our revision. Meanwhile, as stated earlier in this response, we will also add further restrictions in our agreement/license to explicitly prohibit the use that may violate human rights, for both the dataset and the proposed methodology.
>
>
> > The authors state in the questionnaire that they included the estimated hourly wage paid to participants and the total amount spent on participant compensation. I was not able to find this information in the paper and request that the authors include it.
>
> Due to space limit, we moved the annotation cost and time details to Supplementary file Section B `Annotation Cost`. Sorry for the confusion and we will add it back to the main paper given the extra page in the final version.

---

### Review · Ethics_Reviewer_CDAp · 2021-08-11

**Recommendation:**

I believe that a paper that is introducing a data set should provide more context for that data set than is given in this current version of the manuscript.  More information on summary statistics should be provided in the appendix in the usage section that would allow researchers to better have a sense of what the data set could be used for and suggestions for what settings the data should be applicable.  Alternatively, a website could be constructed that outlines the data and provides usage suggestions.  Overall, I think that the data set will be useful for researchers, but the authors could add more context for the data.

**Ethical Issues:**

Yes

**Ethics Review:**

Overall this is a complicated paper because it claims to make three contributions: “(i) We collect the QVHIGHLIGHTS dataset with over 10,000 videos, annotated with human-written natural language queries, relevant moments, and saliency scores. (ii) We propose Moment-DETR to serve as a strong baseline for our dataset. With weakly supervised pretraining, Moment-DETR substantially outperforms several baselines. (iii) We present detailed dataset analyses, model ablations and visualizations.”

The primary issue pointed out by a reviewer is that there is insignificant discussion in the paper on the potential impacts of introducing this data set.  In particular, the authors appear to not have satisfied the checklist for a discussion of potential negative societal impacts.  I believe that part of this is because the paper is both introducing a data set and also deploying new baseline metrics on this data set, thus severely limiting the space provided.

My biggest suggestion is that, given the paper is introducing a data set, more could be done to present summary statistics about the data set and discussions about connections to outlying issues with this distribution in the usage section of the appendix.  I noticed that this is severely lacking in the paper and appendices.  While the authors discuss licensing, they do nothing to describe settings where this data set would apply and be suitable for different questions.

---

> ### Author Response · Authors · 2021-08-20
> **Author Response (Ethics Reviewer CDAp)**
>
> Thanks for reviewing our work and giving constructive feedback. We respond to your questions/suggestions below:
>
> 1, **Summary statistics**: Thanks for the useful suggestions. Currently, we have presented statistics or analyses on (1) video/moment/query lengths; (2) number of queries/videos/moments; (3) moment center distributions; (4) unique nouns and verbs; (5) annotation quality verification; (6) video domains, etc, in Section 3. In our response to `reviewer uoxL`, we also added annotated highlight score distribution. We will also add more statistics and discussions about distributions, etc., as suggested.
>
> 2, **Dataset usage**: In our response to the reviewers, we mentioned “We also plan to restrict the use of the data for research purposes only by having users sign an agreement”, and we will explicitly mention it should not be used for any purpose that may violate human rights (e.g., surveillance).
>
> For the academic community, the dataset is designed to be used in various settings/tasks:
> * single video moment localization[12,6];
> * weakly supervised moment retrieval (Mithun, et al);
> * video corpus moment retrieval [16] (Escorcia, et al)
> * natural language guided highlight detection
>
> Future work may explore other task formats utilizing this data, as long as it is not restricted by the usage restrictions stated above.
>
> Mithun, et al. Weakly supervised video moment retrieval from text queries. CVPR 2019.
>
> Escorcia, et al. Temporal localization of moments in video collections with natural language. arXiv 2019.
>
>
> 3, **Website**:
> Thanks for your suggestion. We agree with your idea of presenting the dataset on a website, we will create a website as you suggested. For now, we have uploaded the data files and some descriptions regarding dataset details and license, see `README.md` file in our supplementary file. We will add additional details regarding statistics and usages as well in our revision and on our website.

---

> > ### Comment · Ethics_Reviewer_CDAp · 2021-08-28
> > **Responding to ethics comments**
> >
> > Thank you for acknowledging the comments! Provided that the summary statistics are in the manuscript (or appendix), that satisfies one of my major concerns.  Additionally, I believe that the added context on the appropriate settings is useful and will help reduce misuse as future reviewers can easily identify situations where the data set is clearly outside of its intended use (though through their discretion, the data set may still be appropriate obviously). Finally, the README.md should suffice to alleviate my concerns about the data set in lieu of a website.  My primary concern was that there should be a landing page of some sort that outlined many of these issues/concerns beyond digging through an appendix in a conference publication.  I think that this satisfies my major concerns.  I appreciate the authors willingness to respond to these comments.

---

### Decision · Program_Chairs · 2021-09-27

**Decision:**

Accept (Poster)

**Comment:**

The authors responded fully to the reviwers' concerns (new training, new baselines ...) and constructively to the ethics reviews. Following the rebuttal, all reviewers recommend acceptance.

Of course, the authors should update the paper and dataset as stated.

Congratulations!